# First high-resolution tropospheric $NO_2$ observations from the Ultraviolet Visible Hyperspectral Imaging Spectrometer (UVHIS)

Liang Xi [1,2], Fuqi Si [1], Yu Jiang [1], Haijin Zhou [1], Kai Zhan [1], Zhen Chang [1], Xiaohan Qiu [1] and Dongshang Yang [1,2]

[1]Key Laboratory of Environmental Optics and Technology, Anhui Institute of Optics and Fine Mechanics, Chinese Academy of Sciences, Hefei 230031, China
[2]Science Island Branch of Graduate School, University of Science and Technology of China, Hefei 230026, China

*Correspondence to*: Fuqi Si (sifuqi@aiofm.ac.cn)

**Abstract.** We present a novel airborne imaging differential optical absorption spectroscopy (DOAS) instrument: Ultraviolet Visible Hyperspectral Imaging Spectrometer (UVHIS), which is developed for trace gas monitoring and pollution mapping. Within a broad spectral range of 200 to 500 nm and operating in three channels, the spectral resolution of UVHIS is better than 0.5 nm. The optical design of each channel comprises a fore-optics with a field of view (FOV) of 40°, an Offner imaging spectrometer, and a charge-coupled device (CCD) array detector of $1032 \times 1072$ pixels. A first demonstration flight using UVHIS was conducted on 23 June 2018, above an aera of approximate 600 $km^2$ in Feicheng, China, with a spatial resolution of about $25 \times 22$ $m^2$. Measurements of nadir backscattered solar radiation of channel 3 are used to retrieve tropospheric vertical column densities (VCDs) of $NO_2$ with a mean total error of $3.0 \times 10^{15}$ molec $cm^{-2}$. The UVHIS instrument clearly detected several emission plumes transporting from south to north, with a peak value of $3 \times 10^{16}$ molec $cm^{-2}$ in the dominant one. The UVHIS $NO_2$ vertical columns are consistent with the ground-based mobile DOAS observations, with a correlation coefficient of 0.65 for all co-located measurements, a correlation coefficient of 0.86 for the co-located measurements that only circled the steel factory and a slight underestimation for the polluted observations. This study demonstrates the capability of UVHIS for $NO_2$ local emission and transmission monitoring.

## 1 Introduction

Nitrogen oxides ($NO_x$), the sum of nitrogen monoxide (NO) and nitrogen dioxide ($NO_2$), play a key role in the chemistry of the atmosphere, such as the ozone destruction in the stratosphere (Solomon, 1999), and the secondary aerosol formation in the troposphere (Seinfeld and Pandis, 2016). In the troposphere, despite lightning, soil emissions and other natural processes, the main sources of $NO_x$ are anthropogenic activities like fossil fuel combustion by power plants, factories, and road transportation, especially in urban and polluted regions. As an indicator of anthropogenic pollution which leads to negative effects both on the environment and human health, the amounts and spatial distributions of $NO_x$ have attracted significant attention. For example, China has become one of the largest $NO_x$ emitters in the world due to its fast industrialisation, meanwhile China has also experienced a series of severe air pollution problems in recent years (Crippa et al., 2018; An et al.,

2019). Therefore, measuring the NO$_x$ distribution by applying different techniques would benefit the pollutant emission detection and the air quality trend forecast (Liu et al., 2017; Zhang et al., 2019).

Compared to NO, Nitrogen dioxide (NO$_2$) is more stable in the atmosphere. Based on the characteristic absorption structures of NO$_2$ in the ultraviolet-visible spectral range, the differential optical absorption spectroscopy (DOAS) technique has been

applied to retrieve light path integrated densities from different platforms (Platt and Stutz, 2008). Combined the imaging spectroscopy technique, imaging DOAS instruments were developed in recent years to determine the temporal variation and the two dimensional distribution of trace gases. The global horizontal distribution of tropospheric NO$_2$ and other trace gases has been mapped and studied by several space-borne sensors, including SCIAMACHY (Scanning Imaging Absorption Spectrometer for Atmospheric CHartographY; Bovensmann et al., 1999), GOME (Global Ozone Monitoring Experiment;

Burrows et al., 1999), GOME-2 (Munro et al., 2016), OMI (Ozone Monitoring Instrument; Levelt et al., 2006) and TROPOMI (TROpospheric Ozone Monitoring Instrument; Veefkind et al., 2012). The Environmental trace gases Monitoring Instrument (EMI; Zhao et al., 2018; Cheng et al., 2019; Zhang et al., 2020), as the first designed space-borne trace gas sensor in China, was launched on 9 May 2018, on-board the Chinese GaoFen-5 (GF5) satellite. The spatial resolution of most space-borne sensors is coarser than $10 \times 10$ km$^2$, except for that of TROPOMI which is $3.5 \times 5.5$ km$^2$.

To achieve a spatial resolution higher than $100 \times 100$ m$^2$ for investigating the spatial distribution in urban areas and individual source emissions, several researchers have applied imaging DOAS instruments on airborne platforms. The airborne imaging DOAS measurement was first performed by Heue et al. (2008) over the South African Highveld plateau. To retrieve urban NO$_2$ horizontal distribution, Popp et al. (2012), General et al. (2014), Schönhardt et al. (2015), Lawrence et al. (2015), Nowlan et al. (2016), and Lamsal et al. (2017) respectively took measurements in Zürich, Switzerland;

Indianapolis and Barrow, USA; Ibbenbüren, Germany; Leicester, England; Houston, USA; and Maryland and Washington DC, USA. In 2013, an airborne measurement focusing on source emissions was taken in China, over Tianjin, Tangshan and the Bohai Bay (Liu et al., 2015). An inter-comparison study of four airborne imaging DOAS instruments over Berlin, Germany suggests a good agreement between different sensors, and the effectiveness of imaging DOAS in revealing the fine-scale horizontal variability in tropospheric NO$_2$ in urban context (Tack et al., 2019).

Here we present a novel airborne imaging DOAS instrument: Ultraviolet Visible Hyperspectral Imaging Spectrometer (UVHIS), which was designed and developed by Anhui Institute of Optics and Fine Mechanics, Chinese Academy of Sciences (AIOFM, CAS). As a hyperspectral imaging sensor with a high spectral and spatial resolution, UVHIS is designed for operation on an aircraft platform for atmospheric trace gas measurements and pollution monitoring over large areas in a relative short time frame. By using the DOAS technique and geo-referencing, the two-dimensional spatial distribution of

tropospheric NO$_2$ of its first demonstration flight over Feicheng, China is also presented in this paper.

This paper is organised as follows: Section 2 presents a technical description of the UVHIS system, and its preflight calibration results. Section 3 introduces the detailed information of its first research flight over Feicheng, China. Section 4 describes the developed algorithm for the retrieval and geographical mapping of tropospheric NO$_2$ vertical column densities

from hyperspectral data. Section 5 presents the retrieved $NO_2$ column densities, and Section 6 compares the airborne measurements with the correlative ground-based data sets from a mobile DOAS system.

## 2 Instrument Details

### 1.1 UVHIS instrument

UVHIS is a hyperspectral instrument measuring nadir backscattered solar radiation in the ultraviolet and visible wavelength region from 200 to 500 nm. The instrument is operated in three channels with the wavelength ranges of 200-276 nm (channel 1), 276-380 nm (channel 2) and 380-500 nm (channel 3) for minimal stray light effects and highest spectral performance. The main characteristics of UVHIS are summarised in Table 1.

Figure 1 shows the optical bench of channel 3 and those of the other two are similar. The optical design of each channel comprises a telecentric fore-optics, an Offner imaging spectrometer, and a two-dimensional CCD array detector. The Offner imaging spectrometer consists of a concave mirror and a convex grating. The backscattered light below the aircraft is collected by a wide-field telescope with a FOV of 40° in the across-track dimension. After passing through a bandpass filter and a 12.5 mm long entrance slit in the focal plane, the light is reflected and diffracted by a concave mirror and a convex grating. The dispersed light is imaged onto a frame transfer CCD detector which consists of $1032 \times 1072$ individual pixels. For the alignment and slight adjustment of the spectrometer, only the central 1000 rows of pixels are well illuminated in the across-track dimension. In the wavelength dimension, the image covers central 1024 columns of pixels on the CCD detector, whilst the left and right edges are used to monitor dark current. The spectral sampling and spectral resolution of all three channels can be found in Table 1.

To reduce dark current and improve the signal-to-noise ratio (SNR) of the instrument, the CCD detector is thermally stabilised at -20 °C with a temperature stability of $\pm 0.05$°C (Zhang et al., 2017). However, the optical bench is not thermally controlled, because the instrument is mounted inside the aircraft platform which has a constant temperature of 20 °C. The UVHIS is mounted on a Leica PAV-80 gyro-stabilised platform that provides angular motion compensation. A high-grade Applanix navigation system on-board is used to receive position (i.e. latitude, longitude and elevation) and orientation (i.e. pitch, roll and heading) information, which are required for accurate geo-referencing. The UVHIS instrument telescope collects the solar radiation backscattered from the surface and atmosphere through a fused silica window at the bottom of the aircraft. In the case of $NO_2$ measurement, all observations in this study only use channel 3.

### 2.2 Preflight calibration

Spectral and radiometric calibration were performed in the laboratory prior to the flights to reduce errors in spectral analysis. For radiometric calibration, we used an integrating sphere with a tungsten halogen lamp for channel 2 and channel 3. For channel 1, a diffuser plate with a Newport xenon lamp was used for sufficient ultraviolet output. With the help of a well-calibrated spectral radiometer to monitor the radiance of calibration system, the digital numbers from the CCD detectors of

the three channels can be converted to radiance correctly. The uncertainty of absolute radiance calibration of the UVHIS is 4.89 % for channel 1, 4.67 % for channel 2 and 4.42 % for channel 3.

Preflight wavelength calibration was also performed in the laboratory, using a mercury–argon lamp and a tuneable laser as light sources. We modelled the slit function of the UVHIS using a symmetric Gaussian function. Spectral registration and slit function calibration were achieved by least square fitting of the characteristic lines in the collected spectra. Table 2 lists the retrieved full-width at half maximum values (FWHMs) for channel 3. Figure 2 shows the measured slit functions at 450.504 nm for nine viewing angles (i.e. -20°, -15°, -10°, -5°, 0°, 5°, 10°, 15°, 20°) and the respective retrieved slit function shapes using a symmetric Gaussian function. These Gaussian fit results suggest that a symmetric Gaussian function is a reasonable assumption for the slit shape in all viewing directions.

## 3 Research flight

The first demonstration flight over Feicheng City, Shiheng Town and neighbouring rural areas was conducted on 23 June 2018, aiming at producing tropospheric $NO_2$ field maps of a large area in a relatively short time frame. Feicheng is a county-level city in Shandong province, approximately 410 km away from Beijing. Figure 3 shows the TROPOMI $NO_2$ tropospheric observation on 23 June 2018, with the background Google map and the location of Feicheng. The flight area is located on the south bank of the Yellow River, at the western foot of Mount Tai. The UVHIS was operated from the Y-5 aircraft at an altitude of 3 km above sea level which is higher than the height of planetary boundary layer (PBL), with an average aircraft ground speed of 50 m/s. An overview of the observation area and the flight lines are provided in Fig. 4. The aircraft took off at local noon from the airfield in Pingyin County, approximately 19 km northwest of the centre of the field. An area of approximately 600 $km^2$ was covered in 3 h, under clean sunny and cloudless conditions with low-speed southerly winds.

The research flight included 13 parallel lines in the east-west direction, starting from the lower left corner in Fig. 2. The distance between adjacent lines was 1.5 km, whilst the swath width of each individual line was approximately 2.2 km. Gapless coverage between adjacent lines can be guaranteed in this pattern because of the adequate overlap. To validate the $NO_2$ column densities retrieved from the UVHIS by comparison to ground measurements, mobile DOAS measurements were taken inside the research area on the same day. As shown in Fig. 4, the measurements of the mobile DOAS system circled around the steel factory and the power plant which are the presumed major emission sources inside the observation area.

## 4 Data processing chain

The $NO_2$ tropospheric vertical column density (VCD) retrieval algorithm of the UVHIS consists of four major steps. First, some necessary pre-processing procedures are required before any spectral analysis of the UVHIS data. Next, the UVHIS spectral data after pre-processing are analysed in a suitable wavelength region by applying of the well-established DOAS

technique. Then, the air mass factors (AMFs) are calculated for every observation based on the SCIATRAN radiative transfer model to convert the slant column densities (SCDs) to tropospheric vertical column densities. In the final step, the geo-referenced $NO_2$ VCDs are resampled and overlaid onto Google satellite map layers.

## 4.1 Pre-processing

The pre-processing procedure before spectral analysis includes data selection, geo-referencing, dark current correction,
spatial binning and in-flight calibration. First, the spectral data acquired during aircraft U-turns are removed in the processing because of the large and changing orientation angles. Furthermore, a radiance threshold of 12.8 µW $cm^{-2}$ $sr^{-1}$ $nm^{-1}$ at 450 nm is set to neglect some over-illuminated ground pixels inside the flight area, which are usually caused by the presence of clouds or water mirror reflection. During the entire flight, the sun glinted on water several times in the southern part of the flight area, especially above the river near the reference area. However, cloud was not present due to the clean
clear-sky weather condition.

Accurate geo referencing is essential for emission source locating and data comparison, and can be achieved with the sensor position and orientation information recorded by the navigation system and IMU on-board.

Dark current correction is performed based on the measurement at the start of the entire flight by blocking the fore-optics, which is necessary to improve the instrument performance and reduce the analysis error in DOAS fit.

In order to increase the SNR of the instrument and the sensitivity to $NO_2$, the raw pixels of the imaging DOAS are usually aggregated in the across- and along-track directions. According to photon statistics when only shot noise is considered, the SNR should rise with the square root of the number of binned spectra. However, this improved SNR of the instrument results in reduced spatial resolution. In the data analysis of the Feicheng flight, we use the binning of 10 pixels in the across-track direction, resulting in a ground pixel size of approximately $25 \times 22$ $m^2$.

Given that the wavelength-to-pixel registration and the slit function shape of the UVHIS could change compared to laboratory calibration results, in-flight wavelength calibration is essential for the next DOAS analysis. This in-flight wavelength calibration is achieved by fitting the measured spectra to a high-resolution solar reference (Chance and Kurucz, 2010) with slit function convolution and wavelength shift. The nominal wavelength-to-pixel registration determined in laboratory calibration, is used as initial values in the iteratively fitting procedure for convergence to the optimal solution. The
effective shifts and FWHMs of different across-track positions are plotted in Fig. 5. The results at three wavelengths are presented as follows: blue for 430 nm (the start of the analysis wavelength region), green for 450 nm (the middle of the analysis wavelength region) and red for 470 nm (the end of the analysis wavelength region).

## 4.2 DOAS analysis

After pre-processing, the observed UVHIS spectra are analysed using the QDOAS (Danckaert et al., 2020) software to
retrieve the $NO_2$ slant column densities. The basic idea of the DOAS approach is to separate broadband signals like surface reflectance and Rayleigh scattering, and narrow-band signals like trace gas molecular absorption. The fit window is 430–470

nm, considered to contain strongly structured $NO_2$ absorption features and with low interference of other trace gases such as $O_3$, $O_4$, and water vapor. The absorption cross-sections of $NO_2$ and other trace gases and a synthetic Ring spectrum are simultaneously fitted to the logarithm of the ratio of the observed spectrum to the reference spectrum. These cross sections are made by convolving the high-resolution cross sections with the in-flight wavelength calibration results for all across-track positions. Further details of the DOAS analysis setting can be found in Table 3.

For each analysed spectrum, the direct result of the DOAS fit is the differential slant column density (dSCD) which is the $NO_2$ integrated concentration difference along the effective light path between the studied spectrum and the selected reference spectrum ($SCD_{ref}$). Reference spectra were acquired over a clean rural area upwind of the urban and factory areas, in the lower left corner of Fig. 4. In the quite homogeneous background area, several spectra were averaged to increase the SNR of the reference spectrum. To avoid across-track biases, a reference spectrum is required for each across-track position because of its intrinsic spectral response. According to TROPOMI tropospheric $NO_2$ product of the reference area on the same day, the residual $NO_2$ amount in the background spectra is estimated to be $3 \times 10^{15}$ molec $cm^{-2}$. Changes in the stratospheric $NO_2$ could also propagate to the measured tropospheric columns of UVHIS. Under the assumption of a constant stratosphere in time and space during the flight, the changes in the SZA impact the column difference between the measurement and the reference. To correct the change in the stratospheric $NO_2$ SCD, we apply a geometric approximation of the stratospheric AMF with a stratospheric VCD of $3.5 \times 10^{15}$ molec $cm^{-2}$ from TROPOMI product. The maximum change in the stratospheric SCD with respect to the reference, was $8 \times 10^{14}$ molec $cm^{-2}$.

A sample $NO_2$ DOAS fit result and the corresponding residual of UVHIS spectra are illustrated in Fig. 6, with a dSCD of $4.95 \pm 0.34 \times 10^{16}$ molec $cm^{-2}$ and a RMS on the residuals of $4.27 \times 10^{-3}$.

### 4.3 Air mass factor calculations

SCD is the integrated concentration along the effective light path of observation, which is strongly dependent on the viewing geometry and the properties that influence radiative transfer of light through the atmosphere. VCD is the integrated concentration along a single vertical transect from the Earth's surface to the top of the atmosphere, which is independent of the changes in the light path length of the SCD.

$$VCD_i^t = \frac{dSCD_i + dSCD_i^s + SCD_{ref}^t}{AMF_i^t} = \frac{dSCD_i + dSCD_i^s + VCD_{ref}^t \times AMF_{ref}^t}{AMF_i^t}. \tag{1}$$

As shown in Eq. (1), the $dSCD_i$ from the DOAS fit can be converted to tropospheric $VCD_i^t$ by dividing the $AMF_i^t$ which accounts for the enhancements in the light path (Solomon et al., 1987). The $dSCD_i^s$ is the stratospheric SCD difference between the measurement and the reference, the $SCD_{ref}^t$, the $VCD_{ref}^t$ and the $AMF_{ref}^t$ are the tropospheric SCD, VCD and AMF of the reference. In this study, tropospheric $NO_2$ AMFs have been computed using the SCIATRAN (Rozanov et al., 2014) radiative transfer model (RTM). The SCIATRAN model numerically calculates AMFs based on a priori information

on the parameters that change the effective light path, such as sun and viewing geometry, trace gas and aerosol vertical profiles and surface reflectance.

### 4.3.1 Parameters in RTM

(1) During flight, the viewing geometry is retrieved from the orientation information of the aircraft. The solar position defined by the solar zenith angle (SZA) and the solar azimuth angle (SAA) as well as the relative azimuth angle (RAA) can be calculated, based on the time information and the latitude and longitude position of each observation. (2) Since the flight is performed under a clear-sky condition, the effect of cloud presence can be ignored in AMF computation. (3) The surface reflectance used in AMF calculation is the product of the Landsat 8 Operational Land Imager (OLI) space-borne instrument

(Barsi et al., 2014). The coastal aerosol band (433 to 450 nm) is selected because its bandwidth is relatively narrow, and this band is basically inside the DOAS fitting window (Vermote et al., 2016). (4) Since no accurate trace gas vertical profile is available during flight, a well-mixed vertical distribution (box profile) of $NO_2$ in the PBL is assumed. However, accurate PBL height is also unavailable, so the typical height of 2 km is a reasonable guess in a sunny summer day in the mid-latitude area in China. (5) The aerosol optical depth (AOD) information used in AMF calculation is the MODIS AOD product

MYD04 at 470 nm on the same day with resampling for every ground UVHIS pixel (Remer et al., 2005), because ground-based aerosol measurement is not performedand no AERONET station data near the flight area is available. The MODIS AOD measurements inside the flight area ranges from 0.14 to 0.36. Like the $NO_2$ profile, the aerosol extinction box profile is constructed from the PBL height and the AOD. A single scattering albedo (SSA) is assumed to be 0.93, and an asymmetry factor is assumed to be 0.68 for aerosol extinction profile, based on previous studies of typical urban/industrial aerosols (Li

et al., 2018).

The Landsat 8 surface reflectance is retrieved through atmospheric correction, using the Second Simulation of the Satellite Signal in the Solar Spectrum Vectorial (6SV) model (Vermote et al., 1997). Since no overpass on the same day existed inside the UVHIS research flight area, we selected the surface reflectance product on 3 May 2018, considering the sunny weather condition and no cloud presence. The spatial resolution of Landsat is approximately 30 m, which is slightly larger

than that of the UVHIS. A resampling of the Landsat 8 surface reflectance product based on nearest neighbour interpolation was performed for every UVHIS ground pixel.

The radiative transfer equation in SCIATRAN is solved in a pseudo-spherical multiple scattering atmosphere, using the scalar discrete ordinate technique. Simulations were performed for the sensor altitude of 3 km above sea level, and the wavelength of the middle of the $NO_2$ fitting windows, i.e. 450 nm. A $NO_2$ AMF look-up table (LUT) was computed, with the

different RTM parameter settings provided in Table 4. For each retrieved dSCD, an AMF was linearly interpolated from the LUT based on the sun geometry, the viewing geometry, and the surface reflectance.

### 4.3.2 RTM dependence study

1. AMF dependence on the surface reflectance

As shown in Fig. 7, a time series of computed AMFs are plotted for the research flight on 23 June 2018, as well as the corresponding surface reflectance, solar zenith angles, and relative azimuth angles. Note that only data of nadir observations are plotted for a clear display, and the time gaps between adjacent flight lines can be observed. Despite the great degree of varieties in viewing and sun geometries, the AMFs strongly depend on the surface reflectance. Previous studies reported by Lawrence et al. (2015), Meier et al. (2017) and Tack et al. (2017) suggest a similar conclusion. A sensitivity test was carried out to investigate the impact of surface reflectance on the AMF calculations based on the SCIATRAN model, with varying values of surface reflectance, and the fixed values of other parameters. The results of this test are shown in Fig. 8 (a) and indicate that the relation between the surface reflectance and the AMF is non-linear. Especially when the surface reflectance is below 0.1, the AMF increase with the surface reflectance rapidly.

Generally speaking, the AMF should be higher in the case of a bright surface reflectance because more sunlight is reflected from the ground back to the atmosphere and then recorded by the airborne sensor. Compared to rural areas, urban and industrial areas usually exhibit enhanced surface reflectance and a subsequent increment in AMF. As shown in Fig. 9, the dependency of the AMF on the surface reflectance is very strong. Moreover, a strong variability of the surface reflectance and the AMF can be observed in these areas. Fig. 9 also shows several slight inconsistencies between the UVHIS measured radiance and the Landsat 8 surface reflectance product. For example, the east-west main road looks thinner in Fig. 7 (a) compared to Figs. 7 (b) and (c). This could be explained by the relatively higher spatial resolution performance of the UVHIS and the resampling of Landsat 8 pixels.

2. AMF dependence on profiles

Based on airborne UVHIS retrieval product, the horizontal distribution of $NO_2$ can be detected, but the vertical distribution of $NO_2$ in the atmosphere is unavailable. The assumptions we made for the profile shape of the trace gas and aerosol extinction do not consider the effective variability during research flight which can be expected in an urban area. Focusing on the impact of different profile shapes on the AMF computation, sensitivity tests of two different $NO_2$ profiles which are closer to ground surface were performed: well-mixed $NO_2$ box profiles of 0.5 and 1 km heights. Compared to the box profile of 2 km which is near the estimated height of PBL, the AMFs decreased by an average of 13 % in the case of a box profile of 1.0 km, whilst the AMFs decreased by an average of 22 % in the case of a box profile of 0.5 km.

Depending on the relative position of the aerosol and trace gas layer, the optical thickness and the scattering properties, aerosols can enhance or reduce the AMF in different ways(Meier et al., 2017). If an aerosol layer is located above the majority of the trace gas, the aerosols with high SSA have a shielding effect as less scatter light passes through the trace gas layer, leading to a shorter light path. On the other hand, if aerosols and the trace gas are present in the same layer, the aerosols can lead to multiple scattering effects which extend the light path and result in a larger AMF. According to the

simulations of a well-mixed aerosol box profile of 2 km and a pure Rayleigh atmosphere, AMFs are slightly higher (by approximately 1 %) than those of the pure Rayleigh scenario.

3. AMF dependence on sun and viewing geometries

Figure 7 shows that the effect of sun and viewing geometries on AMFs is very small. Based on a previous study by Tack et al. (2017), the changing SZA have the greatest effect on the AMFs, compared to other sun and viewing geometries. In this study, we also performed an AMF dependence analysis on SZAs and VZAs. The SZA varied from 12.8° to 37.4° during the 3 h research flight, whilst the VZA ranged from 0° to 30° in most cases. As shown in Figs. 8 (b) and (c), the changes in AMF were less than 10% and 7% respectively, when other parameters were set to the mean values. Generally, a larger SZA or a larger VZA could result in a longer light path through the atmosphere and thus a larger AMF.

4. AMF dependence on the analysis wavelength

The dependence of AMF on the analysis wavelength is shown in Fig. 9. The AMF increases with the analysis wavelength. This could be explained by the Rayleigh scattering characteristics. That is, photons at shorter wavelengths are more likely to be scattered than photons at longer wavelengths, leading to reduced sensitivity to AMF at shorter wavelengths. In the DOAS analysis wavelength window of 430-470 nm, the increase in AMF is approximately 2 %.

**4.4 Resampling and mapping**

The geo-referenced $NO_2$ VCDs are gridded to combine overlapped adjacent measurements, with a spatial resolution of 0.0003° × 0.0002°. Corresponding to 27 × 22 $m^2$, the grid size used is slightly larger than the effective spatial resolution of the UVHIS to reduce the number of empty grid cells. All VCDs are assigned to a grid cell based on its centre coordinates, and several VCDs in one grid cell are unweighted averaged. As shown in Fig. 10, the final $NO_2$ VCD distribution map is plotted over the satellite maps layers in QGIS 3.8 software (QGIS development team, 2020).

**5 Results**

The tropospheric $NO_2$ VCD two-dimensional distribution map is shown in Fig. 10 for the research flight on 23 June 2018. With the high performance of UVHIS in spectral and spatial resolution, Fig. 10 shows fine-scale $NO_2$ spatial variability to resolve individual emission sources. In general, the $NO_2$ distribution is dominated by several exhaust plumes with enhanced $NO_2$ concentration in the northwest partthat share a transportation pattern from south to north that is consistent with the wind direction. These sources include a power plant, a steel factory, two cement factories, and several carbon factories. The largest plume with peak values of up to $3 \times 10^{16}$ molec $cm^{-2}$, originated from an emitter inside a steel factory (number 3 in Fig. 10). This dominant plume reaches its peak value outside at a small valley approximately 1 km north of the factory, and was transporting at least 9 km and seems to be continuing outside the flight region. This enhanced level of $NO_2$ may be caused by the terrain factor which contributes to the accumulation of pollution gases.

Numbers 4 to 6 represent other emitters inside the steel factory, whilst the exhaust plumes from numbers 4 and 5 merged with the dominant plume, the plume from number 6 transported to north individually with a peak value of $1.4 \times 10^{16}$ molec cm$^{-2}$. A plume with peak values of $1.5 \times 10^{16}$ molec cm$^{-2}$ was also detected by UVHIS, which seemed to originate from the power plant. Indicated by number 2 in Fig. 10, this power plant is less than 2 km south of the steel factory. Number 1 in Fig. 10 indicates several carbon factories which are located on the left side of the flight area. Several plumes with peak values of $1.6 \times 10^{16}$ molec cm$^{-2}$, gradually merged during transportation downwind. Numbers 7 and 8 in Fig. 10 represent two different cement factories. The peak values of these two plumes are $1.5 \times 10^{16}$ and $1.4 \times 10^{16}$ molec cm$^{-2}$ respectively.

Compared to the industrial areas mentioned above, the pollution levels of the rural areas are much lower due to the lack of contributing sources, ranging from 2 to $6 \times 10^{15}$ molec cm$^{-2}$. The urban area of Feicheng City is located on the right side of the flight area. Figure 11 is an enlarged map of the UVHIS NO$_2$ observations over Feicheng City, with a colour scale that only extends to $7 \times 10^{15}$ molec cm$^{-2}$. The two black lines in Fig. 11 represent the truck roads in this city. S104 is a provincial highway that crosses Feicheng from north to south, whilst S330 crosses Feicheng from east to west.

Due to temporal discontinuity of the flight lines and the dynamic characteristics of the tropospheric NO$_2$ field, artefacts can be observed between adjacent flight lines. Figure 12 shows three flight lines that pass through the steel factory at local times of 13:26 (a), 13:32 (c) and 14:57 (b). Panels (a) and (b) represent the flight lines that cover the same area with a 1.5 h time gap, and panels (a) and (c) represent adjacent flight lines with a 6 min time gap. These flight lines can be divided into three regions: region A covers no NO$_2$ source but is affected by the carbon factories approximately 3 km away; region B covers the steel factory as the dominant NO$_2$ source; region C covers no NO$_2$ source and is not affected by other sources. In these three regions, only region C is temporally consistent with relatively low NO$_2$ columns, whilst a large temporal variety of NO$_2$ VCDs exists in region A and region B because of inconstant emission sources and changing meteorology.

# 6  NO₂ VCD assessment

## 6.1 Uncertainty analysis

The total uncertainty on the retrieved tropospheric NO$_2$ VCDs is composed of three parts: (1) uncertainties in the retrieved dSCDs, (2) uncertainties in reference column SCD$_{ref}$ and (3) uncertainties in computed AMFs. Assuming that these uncertainties originating from independent steps are sufficiently uncorrelated, the total uncertainty of the tropospheric NO$_2$ VCD can be quantified as follows:

$$\sigma_{\mathrm{VCD}_i} = \sqrt{\left(\frac{\sigma_{\mathrm{dSCD}_i}}{\mathrm{AMF}_i}\right)^2 + \left(\frac{\sigma_{\mathrm{SCD}_{ref}}}{\mathrm{AMF}_i}\right)^2 + \left(\frac{\mathrm{SCD}_i}{\mathrm{AMF}_i^2} \times \sigma_{\mathrm{AMF}_i}\right)^2}. \tag{2}$$

The first uncertainty source, $\sigma_{\mathrm{dSCDi}}$, originates from the DOAS fit residuals and is a direct output in the QDOAS software. This dSCD uncertainty is dominated by the shot noise from radiance, the electronic noise from the instrument, the systematic

uncertainties from the cross sections and the errors from wavelength calibration. In this study, spatial binning of 10 pixels is performed to reduce these DOAS fit residuals, with a mean slant error of $4.8 \times 10^{15}$ molec cm$^{-2}$.

The second uncertainty source, $\sigma_{SCDref}$, is caused by the NO$_2$ residual amount in the reference spectra. Since we use the TROPOMI tropospheric NO$_2$ product of the clean reference area as the background amount, the uncertainty of NO$_2$ vertical column is estimated to be $1 \times 10^{15}$ molec cm$^{-2}$ directly from TROPOMI product. A tropospheric AMF of 2.0 and a tropospheric AMF over the reference spectra of 1.8, result in an uncertainty $9 \times 10^{14}$ molec cm$^{-2}$ to the tropospheric vertical column.

The third uncertainty source, $\sigma_{AMFi}$, derives from the uncertainties in the parameter assumptions of radiative transfer model inputs. According to previous studies (Boersma et al., 2004; Pope et al., 2015), $\sigma_{AMFi}$ is treated as systematic and depends on the surface albedo, the NO$_2$ profile, the aerosol parameters and the cloud fraction. (1) The cloud fraction is neglected in this case because the research flight was under cloudless conditions. (2) The results of the dependence tests in Section 4.3.2 suggest that the surface albedo has the most significant effect on the AMF. According to Vermote et al. (2016), the
uncertainty of the LANDSAT 8 surface reflectance product of band 1 is 0.011. (3) According to the sensitivity study performed in Section 4.3.2, the uncertainty related to the a priori NO$_2$ profile shape is lower than 22 %. (4) According to the performed simulations of a pure Rayleigh atmosphere, the uncertainty related to the aerosol state is estimated to be less than 1 %. (5) Because of the high accuracy of the viewing and sun geometries and their low impact on the AMF computation revealed in the previous section, the uncertainty related to the viewing and sun geometries is expected to benegligible.
Therefore, combining all the uncertainty sources in the quadrature, a mean relative uncertainty of 24 % on the $\sigma_{AMFi}$ is obtained.

Based on above discussion, the total uncertainties on the retrieved tropospheric NO$_2$ VCDs of all the observations of the research flight are calculated, typically ranging from $1.5 \times 10^{15}$ to $5.9 \times 10^{15}$ molec cm$^{-2}$, with a mean value of $3.0 \times 10^{15}$ molec cm$^{-2}$.

## 6.2 Comparison to mobile DOAS measurements

In order to compare the UVHIS NO$_2$ VCDs to the ground-based measurements, mobile DOAS observations were performed on 23 June 2018. This mobile DOAS system is composed of a spectrum acquisition unit and a GPS module. The spectrum collection unit consists of a spectrometer, a telescope, an optical fibre, and a workbench. The FOV of this telescope is 0.3°, and its focal length is 69 mm. The spectrometer used is a Maya 2000 Pro spectrometer, with a wavelength range of 290-420
nm and a spectral resolution of 0.55 nm. The zenith-sky observations of the mobile DOAS were adopted for minimal blocking of buildings and trees in this research. The important properties of the mobile DOAS system and its NO$_2$ retrieval approach are shown in Table 5. It is worth noting that the retrieval window in the mobile DOAS observations differs from the one used for the airborne observations.

For better comparison with the UVHIS $NO_2$ observations, assumptions and parameters in the tropospheric $NO_2$ retrieval
method for the mobile DOAS were similarly set to those of the UVHIS. For example, the residual amount of $NO_2$ in the
reference spectra was set to $3 \times 10^{15}$ molec $cm^{-2}$ with an error of $1 \times 10^{15}$ molec $cm^{-2}$; the mobile DOAS observations only
focused on the tropospheric portion of the $NO_2$ columns, assuming that the difference in the stratospheric $NO_2$ columns
between the observed and reference spectra is negligible; the vertical profiles of $NO_2$ and aerosol extinction, albedo, and
aerosol properties in the AMF calculation were similarly set to those of the UVHIS.

Like the uncertainty analysis of the UVHIS $NO_2$ columns, the total uncertainty on the retrieved mobile tropospheric VCD is
composed of three parts: (1) the mean uncertainty on the dSCD of the mobile DOAS is $1.4 \times 10^{15}$ molec $cm^{-2}$; (2) the
uncertainty of reference vertical column is estimated to be $1 \times 10^{15}$ molec $cm^{-2}$. In the case that the tropospheric AMFs of
the measured and reference spectra are very close, this part results in an uncertainty of $1 \times 10^{15}$ molec $cm^{-2}$ to the total
uncertainty; (3) the mean relative uncertainty on the AMF calculation is 22 % by the square root of the quadratic sum of the
individual uncertainties like UVHIS. Combining these uncertainties together, the mean total uncertainties on the retrieved
tropospheric $NO_2$ VCD is $2.1 \times 10^{15}$ molec $cm^{-2}$.

Basically, the route of the mobile DOAS was designed to encircle the power plant and the steel factory which are supposed
to be predominant sources. For comparison, the mobile DOAS observations are first gridded to the same sampling of the
UVHIS pixels. Thern the VCD of the UVHIS $NO_2$ results is extracted for each co-located mobile measurement. An
overview of the mobile DOAS measurements over the UVHIS $NO_2$ layer is shown in Fig. 13. The $NO_2$ distributions of the
mobile DOAS system and the UVHIS exhibit similar spatial characteristics, i.e. low values are in the south of the steel
factory and power plant, and high values are inside the plumes.

Figure 14 (a) shows scatter plots with the VCDs retrieved by the UVHIS on the x-axis and the mobile DOAS VCDs on the
y-axis, for all co-located measurements. The corresponding results of the linear regression analysis are also provided in
Fig.14 (a), with a correlation coefficient of 0.69, a slope of 1.30, and an intercept of $-9.01 \times 10^{14}$. The absolute time offset
between the mobile DOAS and airborne observations can be up to 1 h, indicating that both instruments cannot sample the
$NO_2$ column at certain geolocations simultaneously. As shown in Fig. 14 (b), when only comparing UVHIS VCDs to mobile
measurements that circled the steel factory, the correlation coefficient improved to 0.86. In this case, all mobile
measurements occurred inside the swath of one flight line of aircraft, and the time offset between two instruments shortened
to 15 min. In general, an underestimation of the UVHIS VCDs of increased value can be observed in Figs 14 (a) and (b).
Considering the variability in local emissions and meteorology, it is reasonable that the differences between these two
instruments exist. A sensitivity test of the AMF on the $NO_2$ profile was performed for all co-located measurements, using a
box profile of 500 m. Compared to the box profile of 2 km, the UVHIS AMFs decreased by an average of 17 %, whilst the
mobile DOAS AMFS decreased by an average of 2.7 %. This results suggest that a more realistic profile with the $NO_2$ layer
closer to the ground could improve the slope closer to unity.

# 7 Conclusions

In this paper, we present the newly developed UVHIS instrument, with a broad spectral region ranging from 200 to 500 nm, and a high spectral resolution better than 0.5 nm. The instrument is operated in three channels at wavelength 200 to 276 nm (channel 1), 276 to 380 nm (channel 2), and 380 to 500 nm (channel 3) for minimal stray light effects and the highest spectral performance. The optical design of each channel consists of a fore-optics with a FOV of 40°, an Offner imaging spectrometer and a CCD array detector of $1032 \times 1072$ pixels.

We also present the first tropospheric $NO_2$ retrieval results from the UVHIS airborne observation in June 2018. The research flight over Feicheng, China, covered an area of approximately $30 \times 20$ km$^2$ within 3 h, with a high spatial resolution approximately $25 \times 22$ m$^2$. We first retrieved the differential $NO_2$ slant column densities from nadir observed spectra by applying the DOAS technique to a mean reference spectra over a clean area. Then we converted those $NO_2$ slant columns to tropospheric vertical columns using the air mass factors derived from the SCIATRAN model with the Landsat 8 surface reflectance product. The total uncertainties of the tropospheric $NO_2$ vertical columns range from $1.5 \times 10^{15}$ to $5.9 \times 10^{15}$ molec cm$^{-2}$, with a mean value of $3.0 \times 10^{15}$ molec cm$^{-2}$.

The two-dimensional distribution map of the tropospheric $NO_2$ VCD demonstrates that the UVHIS is adequate for trace gas pollution monitoring over a large area in a relatively short time frame. With the high spatial resolution of the UVHIS, different local emission sources can be distinguished, fine-scale horizontal variability can be revealed, and trace gas emission and transmission can be understood. For the flight on 23 June 2018, the $NO_2$ distribution was dominated by several exhaust plumes which exhibited the same south to north direction of transmission, with a peak value of $3 \times 10^{16}$ molec cm$^{-2}$ in the dominant plume. The comparisons of the UVHIS $NO_2$ vertical columns with the mobile DOAS observations show a good overall agreement, with a correlation coefficient of 0.65 for all the co-located measurements, and a correlation coefficient of 0.86 for the co-located measurements that only circled the steel factory. However, an underestimation of the high $NO_2$ columns of the UVHIS is observed relative to the mobile DOAS measurements.

The high-resolution information about the $NO_2$ horizontal distribution generated from UVHIS airborne data, is unique and valuable compared to those from ground-based instruments and space-borne sensors. In future study, the UVHIS could be applied in the validation of satellite trace gas instruments and in the connection between local point observations, air quality models, and global monitoring from space.

*Data availability*. The datasets in the present work are available from the corresponding author upon reasonable request.

*Author Contributions*. Conceptualisation, F.S.; methodology, Y.J. and H.Z.; software, Z.C.; validation, X.Q. and D.Y.; formal analysis, L.X.; resources, K.Z.; writing—original draft preparation, L.X.; writing—review and editing, F.S.

*Competing Interest*. The authors declare no conflict of interest.

*Acknowledgments*. We would like to thank Thomas Danckaert, Caroline Fayt and Michel Van Roozendael for help on QDOAS software. We are thankful to the following agencies for providing the satellite data: The Sentinel 5 Precursor TROPOMI Level 2 $NO_2$ product is developed by KNMI with funding from the Netherlands Space Office (NSO) and processed with funding from the European Space Agency (ESA). TROPOMI data can be downloaded from https://s5phub.copernicus.eu. Landsat 8 OLI data have been produced, archived, and distributed by the U.S. Geological Survey (USGS). The original Landsat surface reflectance algorithm was developed by Dr. Eric

Vermote, NASA Goddard Space Flight Center (GSFC). Landsat 8 OLI data are available at https://earthexplorer.usgs.gov.

*Financial support.* This research was supported by grants from the National Key Research and Development Program of China (Nos. 2016YFC0200402, 2019YFC0214702).

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

**Table 1:** UVHIS instrument characteristics of three channels.

| Characteristic | Channel 1 | Channel 2 | Channel 3 |
|---|---|---|---|
| Wavelength range | 200–276 nm | 276–380 nm | 380–500 nm |
| Spectral sampling | 0.074 nm | 0.10 nm | 0.12 nm |
| Spectral resolution | 0.34 nm | 0.46 nm | 0.49 nm |
| FOV | 40° | 40° | 40° |
| Focal length | 18 mm | 18 mm | 18 mm |
| Across-track angular resolution | 0.5 mrad | 0.5 mrad | 0.5 mrad |
| f-number | 3.4 | 3.6 | 3.6 |
| Detector size | 1032 × 1072 | 1032 × 1072 | 1032 × 1072 |


**Table 2.** Preflight wavelength calibration results (FWHMs) of UVHIS channel 3 for 9 viewing angles. Light sources used in the calibration are a mercury-argon lamp and a tuneable laser. Slit function shapes are retrieved by least square fitting of characteristic spectral lines, using a symmetric Gaussian function.

| FOV | 379.887 nm | 404.656 nm | 450.504 nm | 500.566 nm |
|---|---|---|---|---|
| -20° | 0.35 nm | 0.35 nm | 0.39 nm | 0.50 nm |
| -15° | 0.33 nm | 0.31 nm | 0.33 nm | 0.43 nm |
| -10° | 0.31 nm | 0.29 nm | 0.29 nm | 0.41 nm |
| -5° | 0.31 nm | 0.30 nm | 0.29 nm | 0.34 nm |
| 0° | 0.31 nm | 0.32 nm | 0.30 nm | 0.30 nm |
| 5° | 0.34 nm | 0.36 nm | 0.34 nm | 0.30 nm |
| 10° | 0.38 nm | 0.39 nm | 0.38 nm | 0.32 nm |
| 15° | 0.40 nm | 0.44 nm | 0.42 nm | 0.35 nm |
| 20° | 0.45 nm | 0.46 nm | 0.47 nm | 0.38 nm |



**Table 3.** Main analysis parameters and absorption cross sections for $NO_2$ DOAS retrieval.

| Parameter | Settings |
|---|---|
| Wavelength calibration | Solar atlas, (Chance and Kurucz, 2010) |
| Fitting interval | 430−470 nm |
| Cross sections | |
| $NO_2$ | 298 K, Vandaele et al. (1998) |
| $O_3$ | 223 K, Serdyuchenko et al. (2014) |
| $O_4$ | 293 K, Thalman and Volkamer (2013) |
| $H_2O$ | 293 K, Rothman et al. (2013) |
| Ring effect | Chance and Spurr (1997) |
| Polynomial term | Order 5 |
| Offset | Order 1 |


**Table 4.** Overview of the input parameters in the SCIATRAN RTM, characterizing the AMF LUT.

| RTM Parameter | Grid settings |
|---|---|
| Wavelength | 450 nm |
| Sensor altitude | 3 km |
| Surface reflectance | 0.01−0.4 (steps of 0.01) |
| Solar zenith angle | 10−40° (steps of 10°) |
| Viewing zenith angle | 0−40° (steps of 10°) |
| Relative azimuth angle | 0−180° (steps of 30°) |
| Aerosol optical depth | 0−1 (steps of 0.1) |
| Aerosol extinction profile | Box of 2.0 km |
| $NO_2$ profile | Box of 2.0 km |


**Table 5.** Properties of the mobile DOAS system and its $NO_2$ fit.

| Parameter | Settings |
|---|---|
| Elevation angle | zenith |
| Fitting interval | 356−376 nm |
| Wavelength calibration | Mercury lamp |
| Cross sections | |
| $NO_2$ | 298 K, Vandaele et al. (1998) |
| $O_3$ | 223 K, Serdyuchenko et al. (2014) |
| $O_4$ | 293 K, Thalman and Volkamer (2013) |
| Ring effect | Chance and Spurr (1997) |
| Polynomial term | Order 5 |
| Offset | Order 1 |





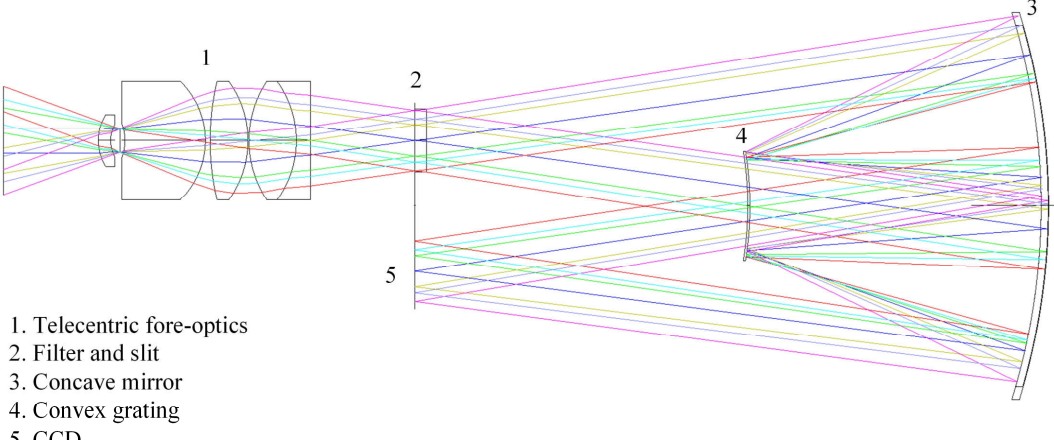

1. Telecentric fore-optics
2. Filter and slit
3. Concave mirror
4. Convex grating
5. CCD

**Figure 1.** Optical layout of the UVHIS channel 3. Optical design of channel 1 and channel 2 is similar.





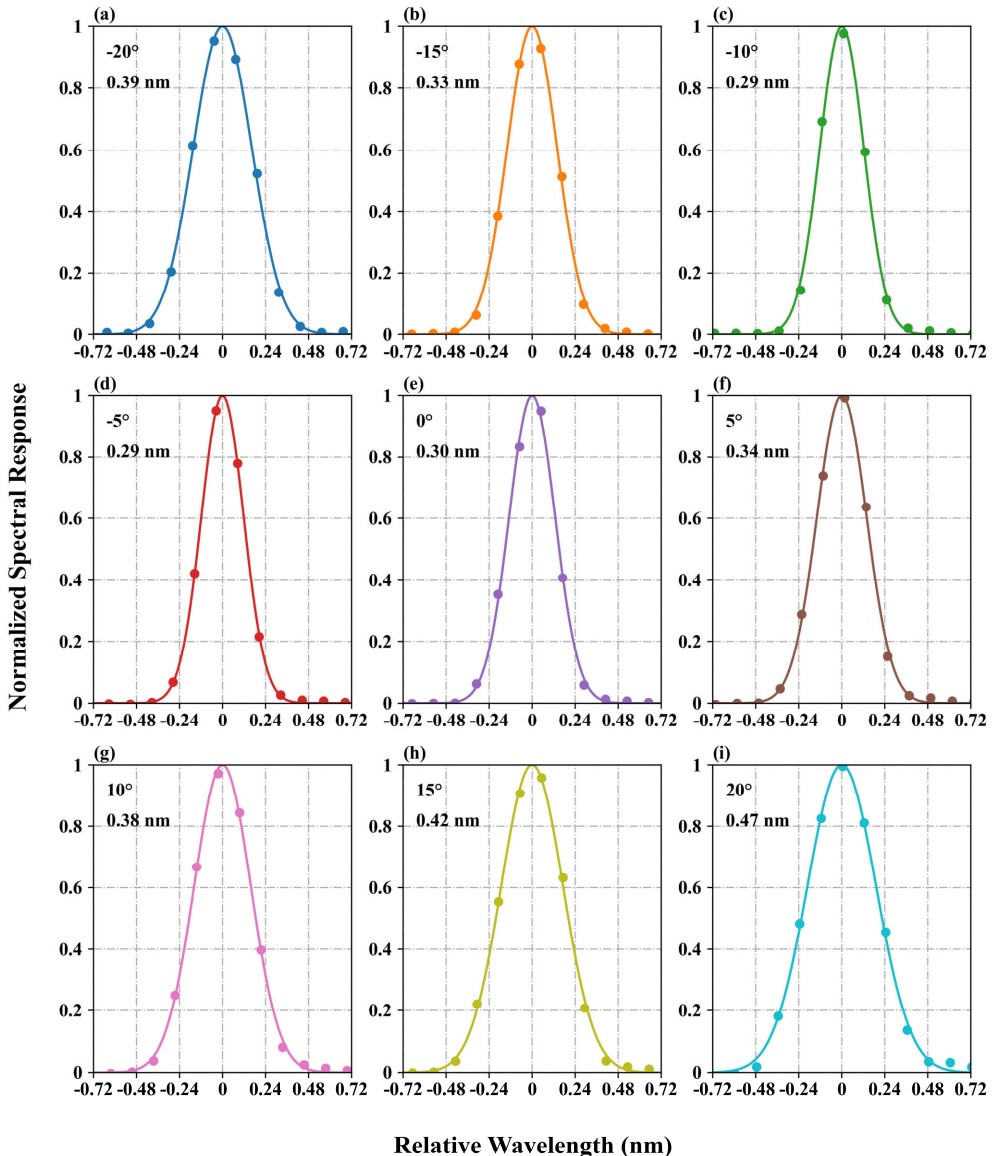


**Figure 2.** Measured slit functions (dots) at 450.504 nm and retrieved slit function shapes (lines) using a symmetric Gaussian function for 9 viewing angles.

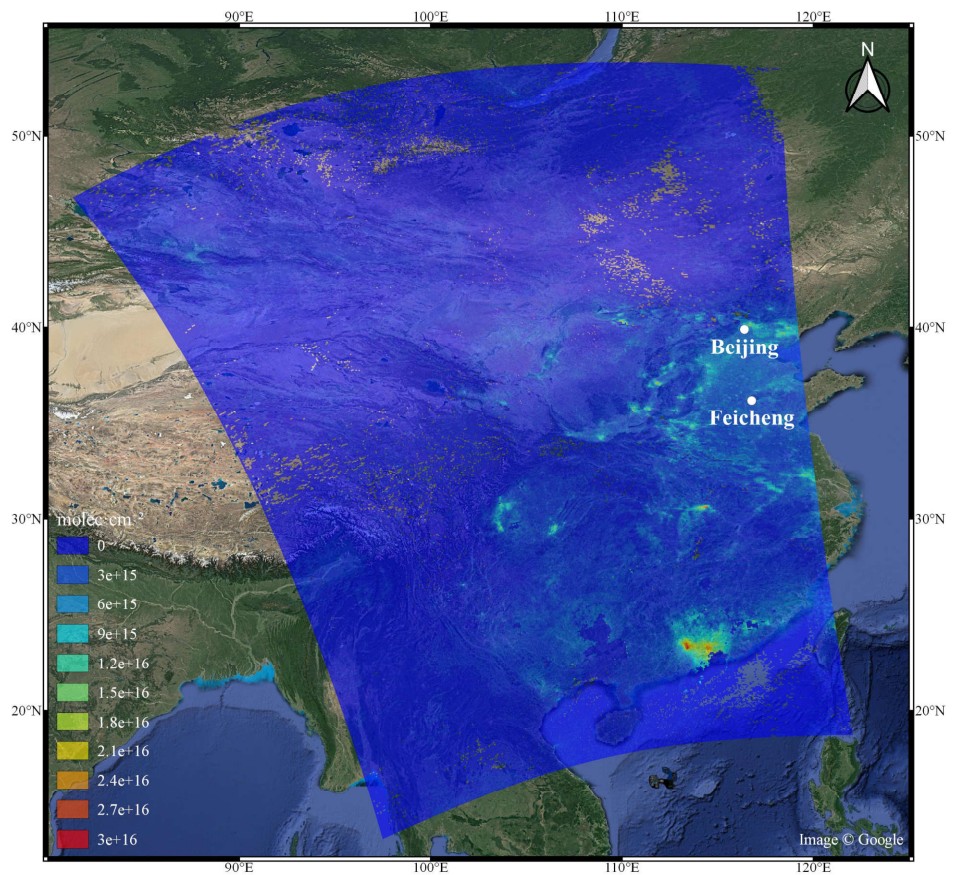

**Figure 3.** TROPOMI observation of tropospheric NO₂ over China on 23 June, 2018. The location of UVHIS flight (Feicheng city) is also

plotted in the map.

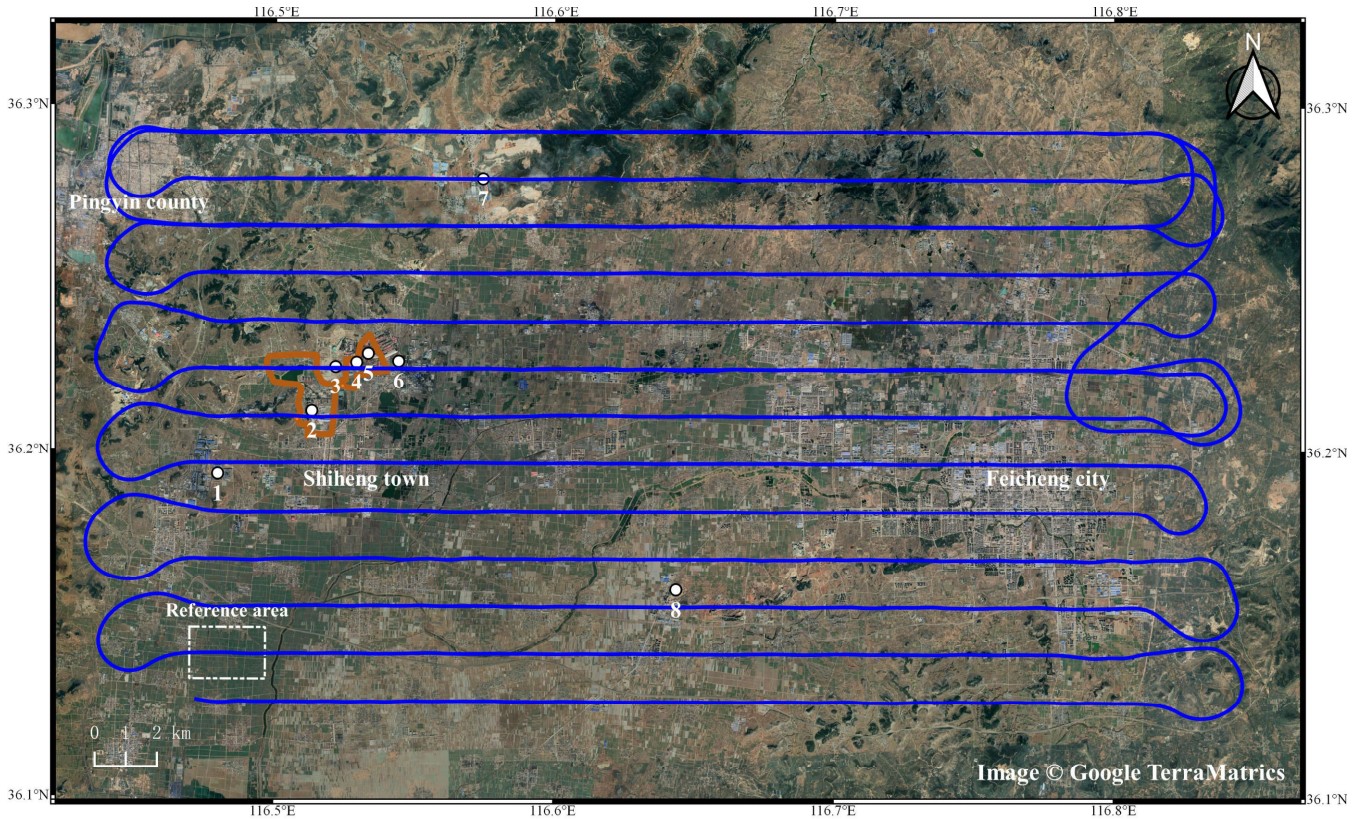

**Figure 4.** Overview of the Feicheng demonstration flight on 23 June, 2018. Flight lines are shown in blue. Two orange circles represent the routes of mobile DOAS system. White dots numbered from 1 to 8 represent the major emission sources. Number 1: several carbon factories; number 2: a power plant; numbers 3-6: individual emitters inside the steel factories, while numbers 4 and 5 are inside the circle of one mobile DOAS route; numbers 7-8: two cement factories. White dashed box represents the reference area.

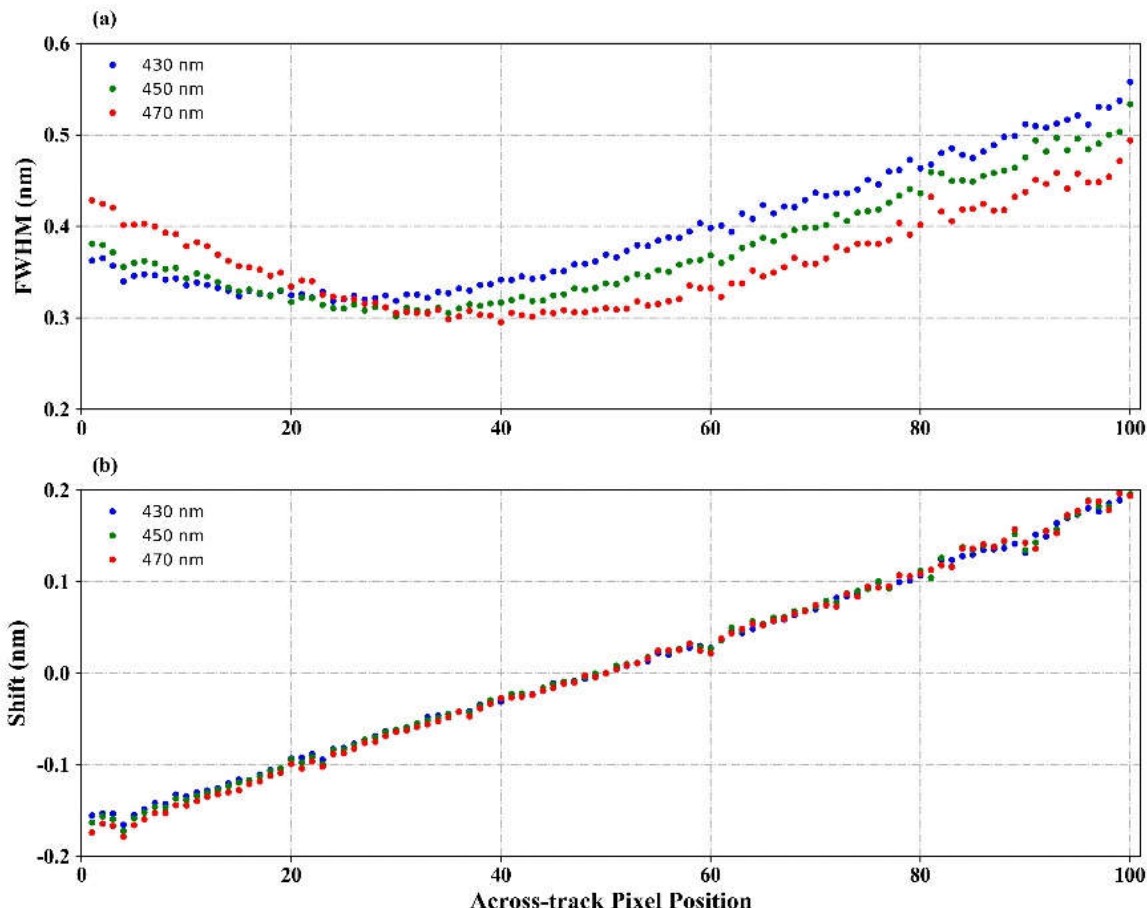

**Figure 5.** In-flight spectral calibration: (**a**) the spectral resolution (FWHM); (**b**) the spectral shift on different across-track position. Results at three wavelengths are plotted: blue for 430 nm, green for 450 nm and red for 470 nm.

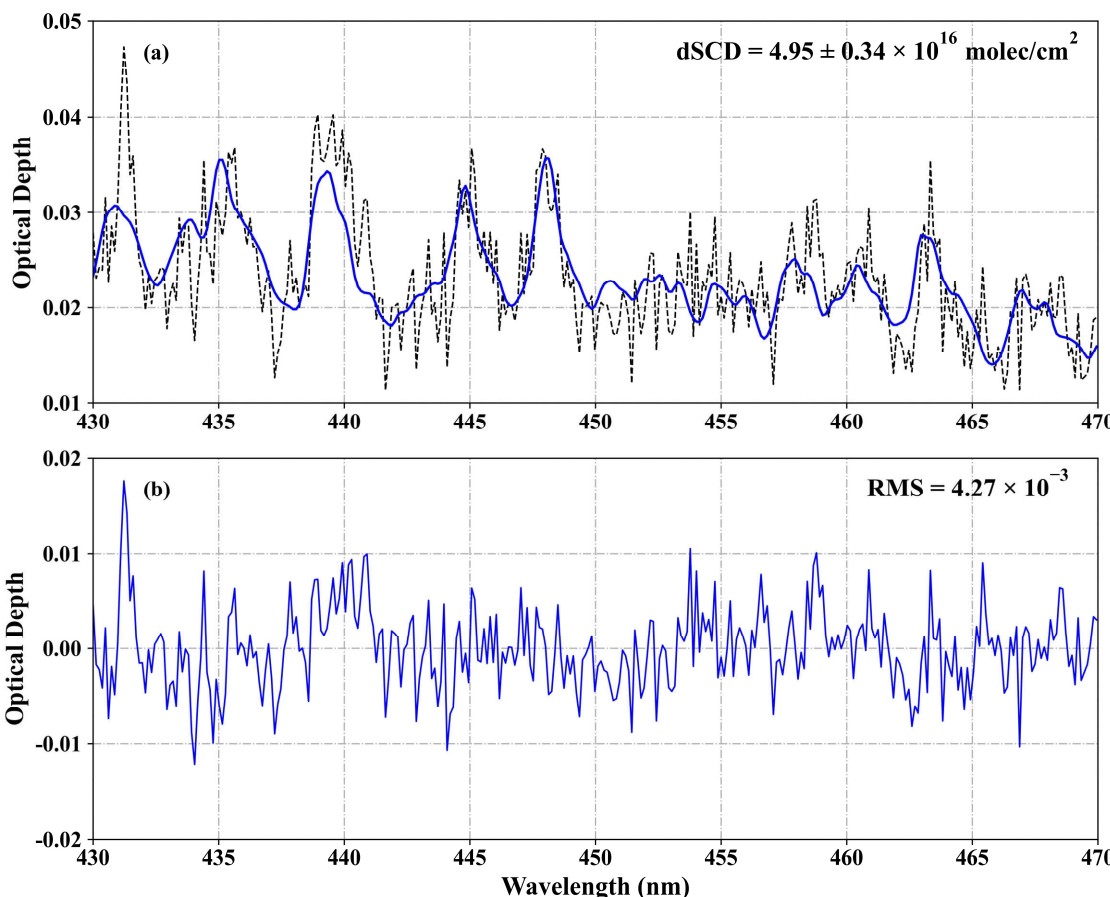

**Figure 6.** Sample DOAS fit result for $NO_2$: (**a**) observed (black dashed line) and fitted (blue line) optical depths from measured spectra; (**b**) the remaining residuals of DOAS fit.

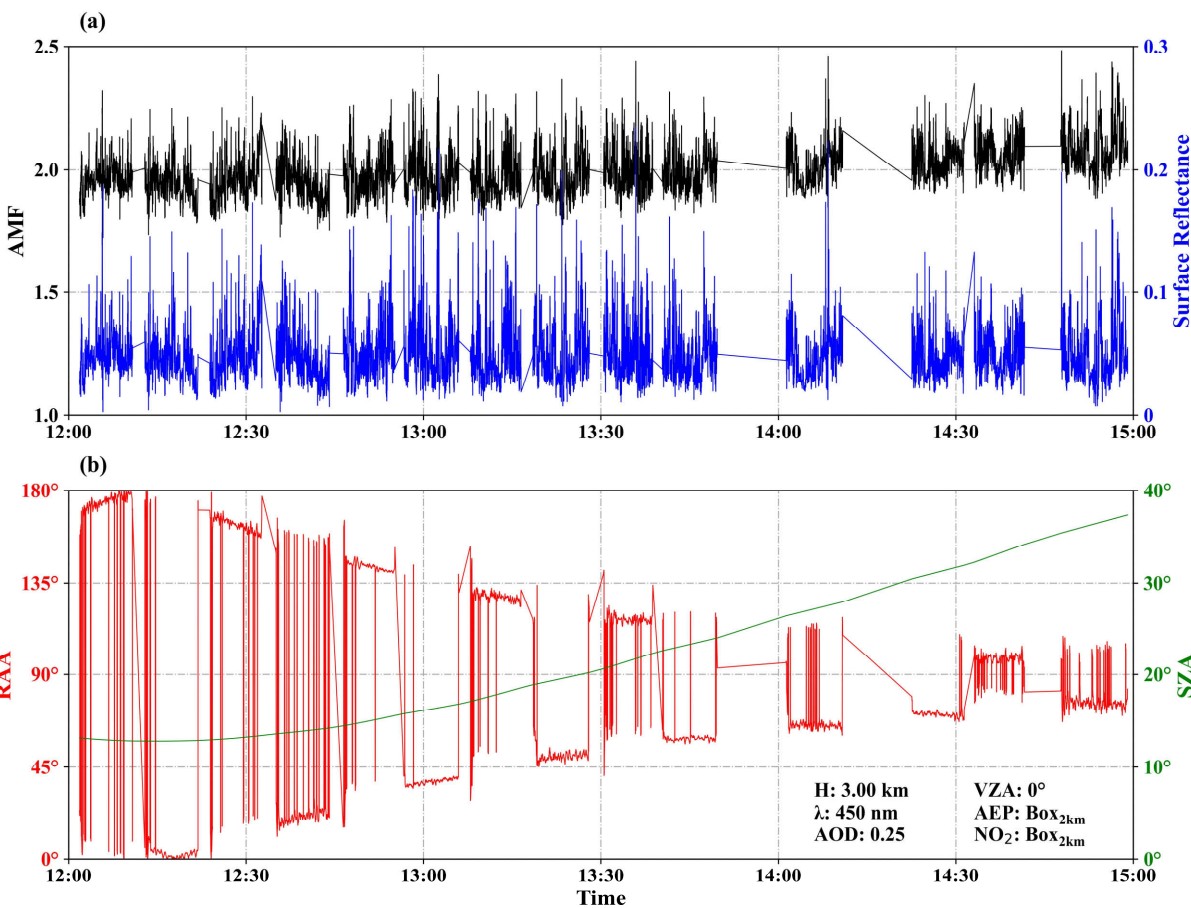

**Figure 7.** Time series of NO₂ AMF compared with **(a)** surface reflectance; **(b)** SZA and RAA for the research flight on 23 June 2018, computed with SCIATRAN model based on the RTM parameters from the UVHIS instrument. Only data of the nadir observations in each flight line are plotted.



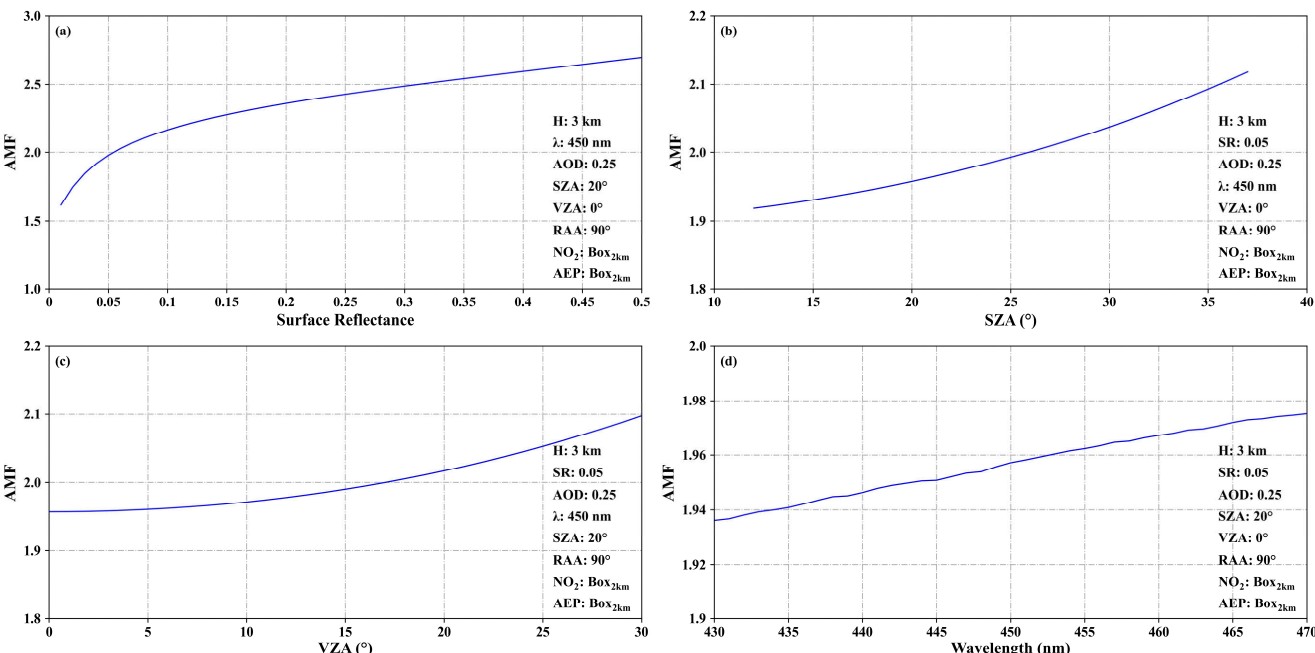

**Figure 8.** AMF dependence analysis results (a): on the surface reflectance; (b): on the SZAs; (c): on the VZAs; (d): on the wavelength.


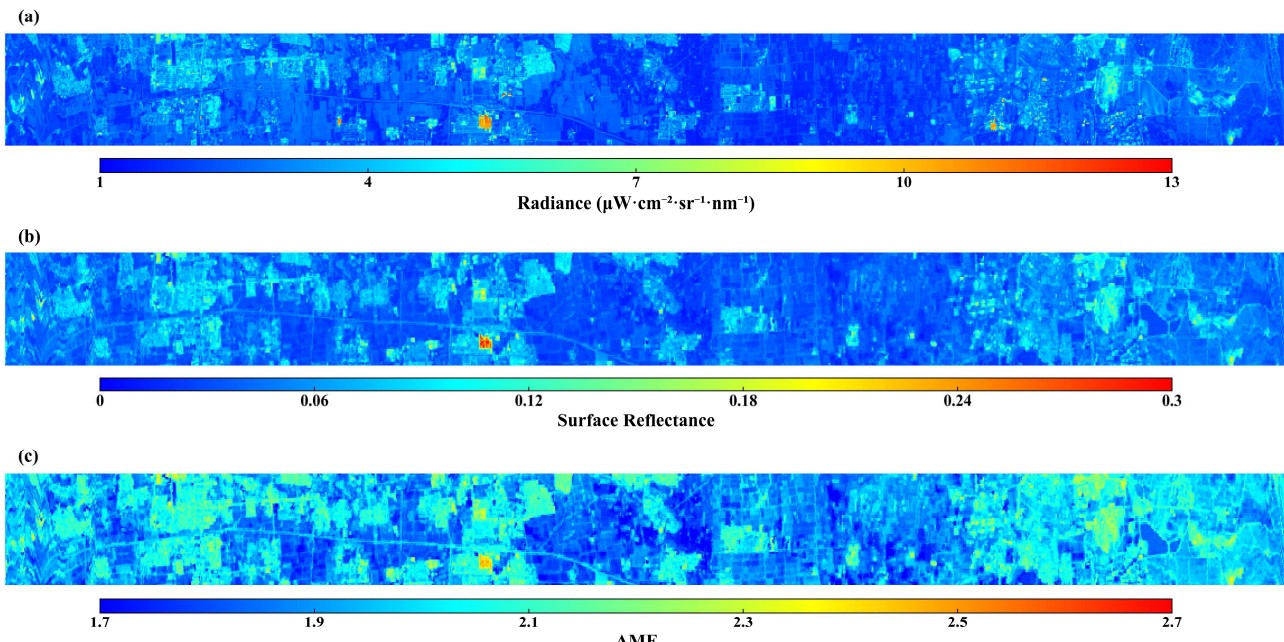

**Figure 9. (a)** UVHIS measured radiance; **(b)** Landsat 8 Surface reflectance; (c) computed AMFs, for one flight line of the Feicheng data set. A strong dependency of the AMF on the surface reflectance can be observed.





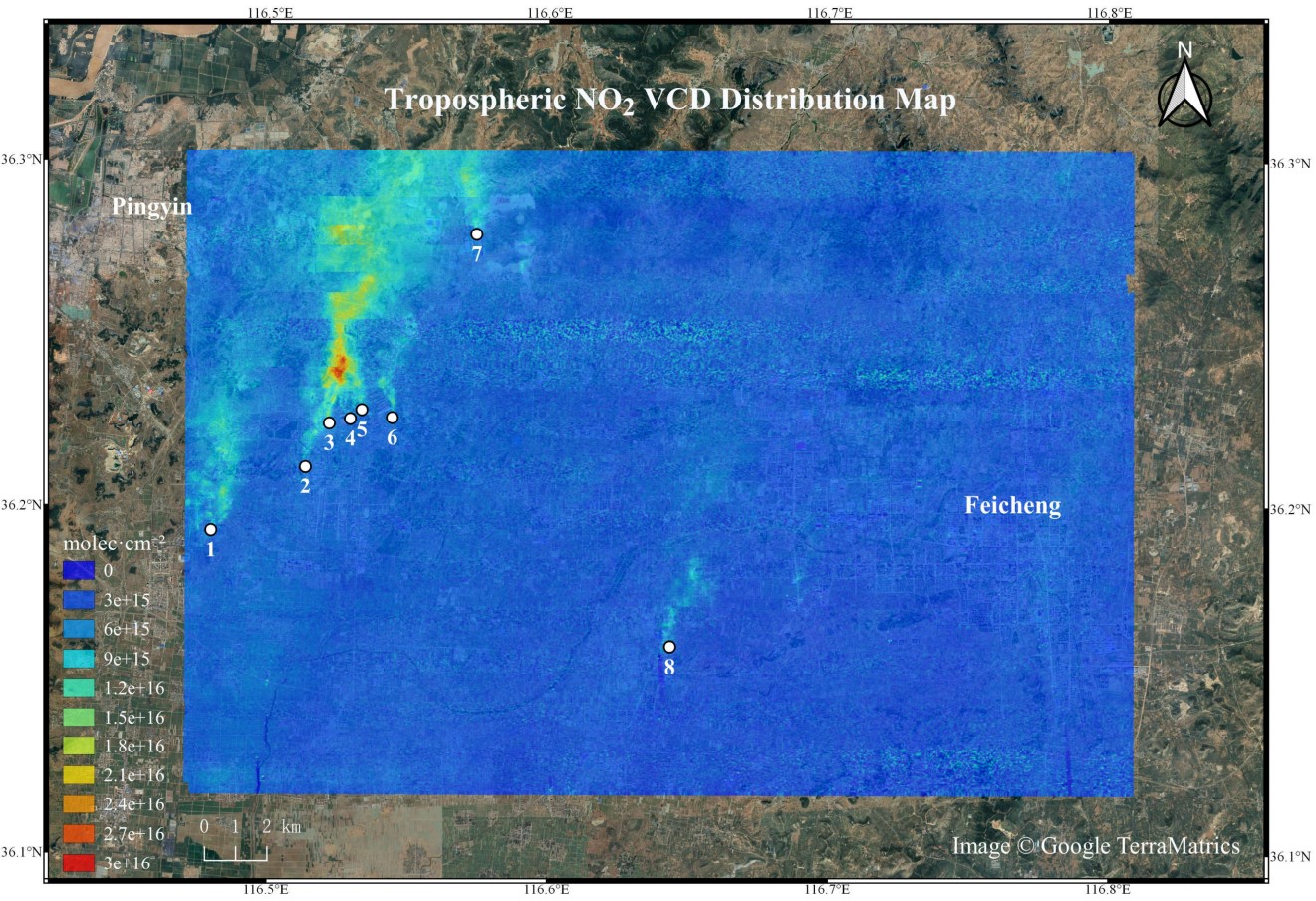

**Figure 10.** Tropospheric NO₂ VCD map retrieved from UVHIS over Feicheng on 23 June 2018. The major contributing NO$_x$ emission sources are indicated by numbers 1 to 8.



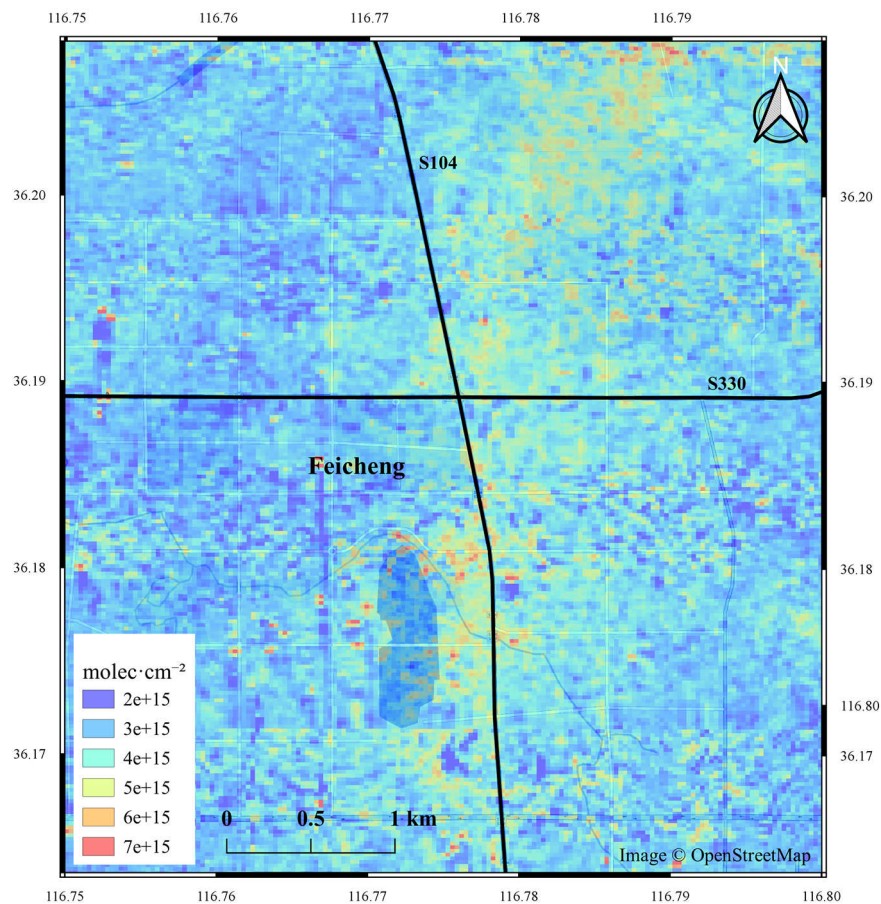

**Figure 11.** Enlargement of UVHIS NO₂ VCD map over Feicheng city with a colour scale only extends to $7 \times 10^{15}$ molec cm$^{-2}$. Two black lines in the map represent two truck roads that cross Feicheng city: S104, and S330.

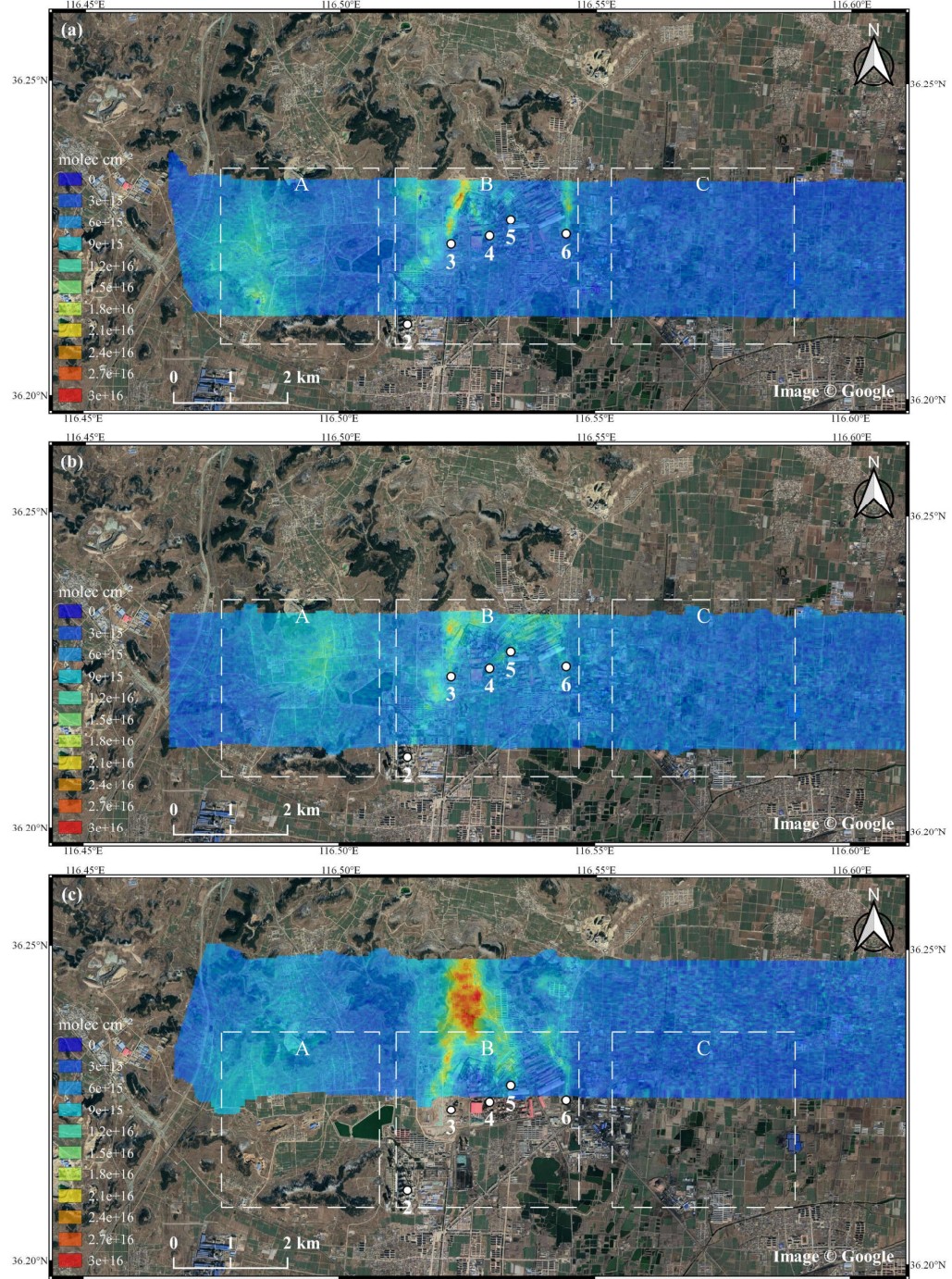

**Figure 12.** Three flight lines that pass through the steel factory, at local time 13:26 (a), 13:32 (c), and 14:57 (b). Panel (a) and (b) represent flight lines that cover the same area with a 1.5 h time gap, panel (a) and (c) represent adjacent flight lines with a 6 min time gap.

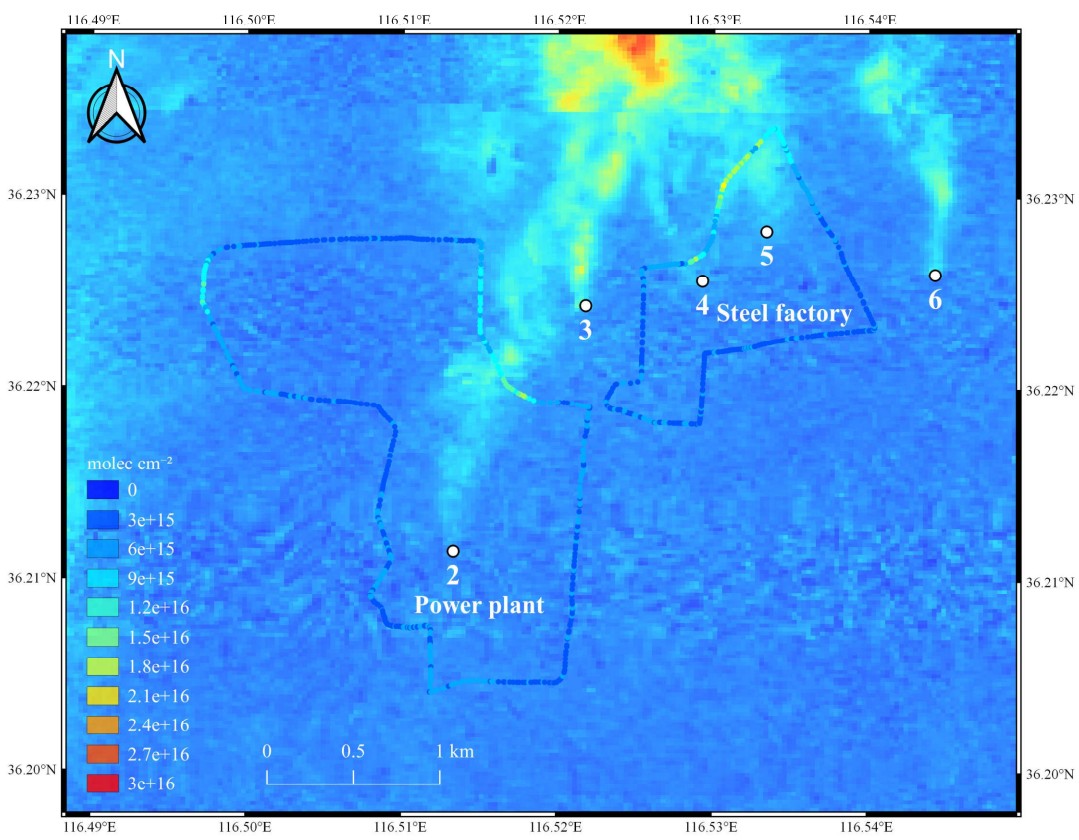

**Figure 13.** Overview of VCDs retrieved from ground-based mobile DOAS system (circle marks), and VCDs retrieved by UVHIS (background layer), measured on 23 June 2018.



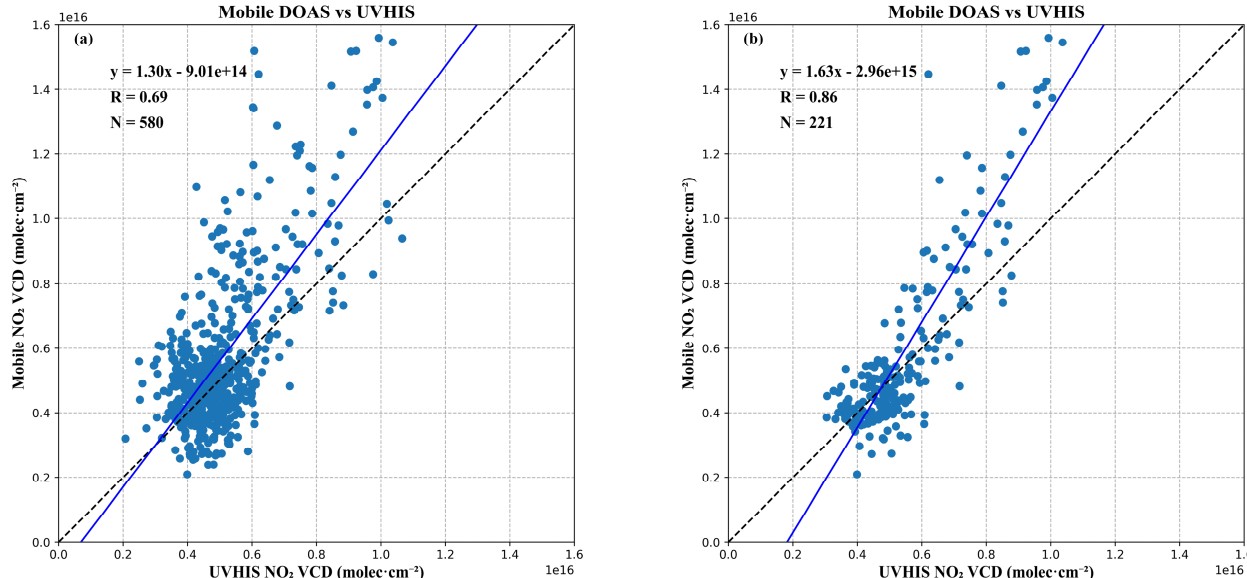

**Figure 14.** Scatter plot and linear regression analysis of the co-located NO₂ VCDs, retrieved from UVHIS and mobile DOAS system, (a) for all co-located measurements, with a time offset of 1 h, (b) for co-located measurements that only circled the steel factory, with a time offset of 15 min.
