# Peer review of "The first high-resolution tropospheric NO2 observations from the Ultraviolet Visible Hyperspectral Imaing Spectrometer (UVHIS)"

_Atmospheric Measurement Techniques, 2020_

## Referee Comment (RC1) · Anonymous Referee #1 · 24 Aug 2020

Comments to amt-2020-225 titled, 'The Ultraviolet Visible Hyperspectral Imaging Spectrometer(UVHIS), and high-resolution NO2 mapping from its first airborne observation'

This manuscript the first measurements from an airborne instrument, UVHIS, which measures backscattered light in the UV and visible parts of the electromagnetic spectrum. It includes instrument characteristics and background, data process-ing/calibration steps, and results from a NO2 VCD retrieval from its research flight near Feicheng, China. This paper fits the scope of AMT and would be a welcome read to the AMT audience for a new capability for high spatial resolution trace gas observations in a new region of the world. However, before publishing, this manuscript requires the

addressing of technical corrections that cause some concern, expansion of details in places, and some improvement on the quality of the writing. While most the comments are minor and related to the writing, I do recommend revisions in the major category because of the concern about the VCD measurement (which may be correct but needs more details to describe to convince the reader) and a great expansion of the mobile DOAS description. Detailed comments on those are below.

Major comments: There are concerns about the actual calculation of the NO2 VCD as described in Sect 4.3. It is hard to tell what the VCD actually represents in the calculation. Is it a total column VCD? If so, the stratospheric details are missed. However, as it is stated that the stratospheric column is assumed stable during the flight from the reference and is canceled out leads the reader to believe it is not a total column. If it's just the below aircraft column, please state this and ensure the proper accounting for the distinction in the AMF and VCD calculation (e.g., Lamsal et al., 2017 is a great example that shows the breakdown of how these components are calculated).

Lamsal, L. N., Janz, S. J., Krotkov, N. A., Pickering, K. E., Spurr, R. J. D., Kowalewski, M. G., Loughner, C. P., Crawford, J. H., Swartz, W. H. and Herman, J. R.: High-resolution NO2 observations from the Airborne Compact Atmospheric Mapper: Retrieval and validation: High-Resolution NO 2 Observations, Journal of Geophysical Research: Atmospheres, 122(3), 1953–1970, doi:10.1002/2016JD025483, 2017.

Line 157-158: Pertaining to the spatial and temporal variability of the stratosphere being stable. This is maybe close to correct for a 3 hours flight, however there are changes in the SZA which will impact the slant column difference between the measurement and the reference. This could be estimated with a geometric calculation of the slant path with an assumed stratospheric amount between the reference and the measurement. It likely is small.

Lines 306 and 288: The mean slant column fitting error of 4.8x1015 molecules cm-2 and mean total value of error of 2.6x1015 molecules cm-2 for the VCD column with a

range going down to 1x1015 molecules cm-2 does not seem to work out mathematically unless the AMF error is zero (which it is not) and the AMF must be  $\sim$ 2 (which is seems to be below that for most cases) and the error of the reference itself is 100% which is stated as 1x1015. Please check this math.

Line 17: the error of 2.6 x1015 is not the fitting error as stated. It is the error based on all sources of uncertainty.

Are there literature references for the mobile-DOAS measurements? If not, then details on the specifics of that measurement need to be greatly expanded upon as well as the zenith-sky NO2 retrieval in Sect 6.2. Especially details on the uncertainty and what the VCD represented vertically (just the troposphere? Stratosphere?).

In Figure 2, there are lines that are repeated in the northern half of the raster. Can this be described? Does this impact the comparisons to the mobile DOAS measurements? Please describe this overlap in the paper is this is what is shown in the NO2 data. The details should be discussed in the paper.

Minor comments:

Line 108: The mobile DOAS measurements are not shown in Figure 5 as stated. However, are shown in Figure 11. Consider adding the location of these measurements in Figure 2 to show where the mobile DOAS measurements were taken. Additionally, in Lines 288-290: technical details about mobile DOAS measurements are not mentioned before this. Discuss these points within Sect 6.2.

Line 28: What is the intended meaning behind 'that NOx attracts large attention'. Please elaborate with some details and examples.

Line 54: add 'of NO2' after spatial distribution to clarify that this is the gas of interest in this paper.

Lines 67-68. Figure 1 only shows the optical bench for one of the channels and not all three as implied by the text. Please fix to the text saying that Figure 1 shows the optical

bench for channel 3 and that the other two are similar.

Line 86-87: reword this sentence to say that spectral and radiometric calibration in the laboratory were done prior to flights to reduce errors in spectral analysis. There shouldn't be a need to state it as 'very necessary'.

Line 38: Is this the first space-borne sensor ever in China or the first space-borne sensor related to air quality or trace gases?

Line 126: clouds were mentioned as filtered out. However, in the rest of the paper it says that the conditions were cloud free. Were there clouds to be filtered? If so, state where and how cloudy it was. If not, state that cloud filtering was not needed for these measurements due to clear skies. Same comment with the sun glint on water if applicable.

Line 165: Please revise to say something like 'and the properties that influence radiative transfer of light through the atmosphere' instead of 'and the radiative transfer'

Line 184: Please clarify which MODIS AOD product was used.

Line 184: What was the AOD measured from MODIS was during the flight? Please add this detail into the manuscript.

Line 187: Please justify why SSA of 0.93 and asymmetry factor of 0.68 are used.

Line 226: Is [28] a referring to a reference? Please fix.

Consider consolidating Figures 6, 8, and 9 into one figure.

Line 258: The difference in adjacent flight lines are not 'biases' but rather 'artifacts' of the changing NO2 VCDs due to temporal variation.

Section 6 would benefit from a more descriptive title, like 'NO2 VCD Assessment' rather than 'Discussion'

Lines 324-325. How do these results change if only considering points with a stricter

temporal window between the mobile and aircraft measurements?

In line 323, the difference between the mobile DOAS measurements and the airborne measurements is described as an 'overestimate' of mobile DOAS measurements, but in the conclusions and abstract it is stated as an 'underestimate' by the aircraft. Please be consistent in this description in the manuscript.

Figure 2: the black dots are hard to see. Please change the color and/or symbol to make the points of interest stand out.

Instead of having Table 2, could those results be translated into Figure 3(a) somehow? If keeping Table 2, then be more descriptive in the caption to say these are FWHMs at these wavelengths/angles.

Lines 110-113: This text is redundant. These details were already stated in the previous paragraph.

Similarly, the first two paragraphs in Section 5 appear to be redundant. Please consolidate into one paragraph without repeating details already stated.

There are grammar mistakes throughout the manuscript. These errors will need to be fixed before publication but I expect will be evolving in revisions. Some grammar and other writing fixes are located at the bottom of this review to help gives examples as to the types of errors found. They are not a full edit.

Below are non-mandatory comments to address, but suggestions that could be interesting to investigate if the authors were up for adding these details:

Consider a more concise title, such as, 'The first high-resolution NO2 observations from the Ultraviolet Visible Hyperspectral Imaging Spectrometer (UVHIS)'.

Does EMI capture this area? Or TROPOMI? It would be interesting to show some comparisons to those data products, especially since the flight was early afternoon on a cloud free day.

What does Feicheng City look like if mapped on a color scale that only extends to 5x1015. Are there spatial patterns captured? It is hard to see any patterns in Figure 10 in that area due to the color scale expanding to much larger pollution scales. Perhaps a second panel in this figure would be interesting.

Figure 7: consider adding a true color image of this line to compare with the surface reflectance and AMF.

Can you comment on applications of the other channels for UVHIS? Are there plans for other products in the future?

Writing edits: Line 31: delete 'well' Line 36: Delete 'and' Line41: change 'for investigation of' to 'to investigate the' Line 42: 'researches' should be 'researchers' Line 64: change 'at wavelength' to 'with the wavelength ranges of' Line 77: insert 'the' between 'improve' and 'signal-to-noise' Line 77: insert 'the' before 'CCD' Line 79: delete 'all' Line 79: 'basically temperature consistent' is casual more than technical. What is the temperature consistency? Line 101: it should be flight lines 'are' provided Line 108: Delete 'basically' Line 116: Add 'The' in front of 'NO2 vertical column density' Line 130: delete 'and across-track direction' Line 154: it should be Reference spectra 'were' acguired Line 156: change 'detector' to 'position' Line 173: vertical profiles, and surface reflectance. (delete etc.) Line 185: data near the flight area 'are' available Line 195: AMFs 'are' plotted for 'the' research flight Line 206: Change 'industry' to 'industrial' Line 212: Results of this test 'are' shown in Fig. 6, 'and' indicate Line 226: delete 'the' Line 227: less scattered light 'passes' through (recommend deleting solar, too). Line 233: Based on 'a' previous study 'from' Tack Line 234: delete 'the' before 'changing' Line 234: change 'among other' to 'in comparison to' Line 239: Delete 'It is obvious that the' and change 'increase' to 'increases' Line 240: shorter wavelengths are more 'likely to be' scattered Line 241: delete 'before NO2 layer' Line 273: delete 'the' Line 318: delete 'It is obvious that'

---

## Referee Comment (RC2) · Anonymous Referee #2 · 7 Sep 2020

Referee comment to: The Ultraviolet Visible Hyperspectral Imaging Spectrometer (UVHIS), and high-resolution NO2 mapping from its first airborne observation by Xi, L., Si, F., Jiang, Y., Zhou, H., Zhan, K., Chang, Z., Qiu, X., and Yang, D

General comment:

Xi et al. present a new airborne imaging DOAS instrument and results of the first demonstration flight. The results are encouraging and data might be interesting for further analysis such as satellite validation, emission estimates or model comparison. The paper fits very well in the context of AMT. However at some points more details might be required by the reader. Most of them are not critical but require an update of

the manuscript.

Major Comments:

Through out the manuscript the authors should take care to distinguish between total and tropospheric vertical columns - in most cases the tropospheric vertical column or to be more precise the column below the flight altitude is meant e.g. I 167.

The NO2 fit shown in figure 4 has some residual structures, which might be noise but might also be caused by a systematic issue. The instrument was carefully calibrated before the measurements. The wavelength calibration is used only as apriori for the QDOAS software - which is certainly necessary. The slit function shape that can be extracted from the measurements using the Mercury-Argon lamp but are not shown or mentioned. Instead a symmetric Gaussian slit function is assumed in the DOAS analysis. The width of the slit function varies significantly within the fitting window (figure 3) - maybe the shape does so as well? I suggest adding at least a figure of the measured slit function for the extreme viewing directions (left, centre and right) including the Gaussian fit.

The observations partly overlap as the distance between the parallel flightracks was 1.5 km and the swath width is 2.2 km. How good do the observed tropospheric NO2 columns agree in the overlapping regions, does this depend on the flight direction and time, according to figure 2 some parts of the flight track were covered at least twice.

Minor Comments:

Abstract:

The error given here is not the fitting error but the total error of the VCD - even if the fitting error  $\sim$ 4.8 x 1015 molec/cm2 is probably the dominant contribution.

Section 2 instrumental Details:

The instrument was build for airborne measurements in the troposphere, however the
spectral range encompasses the deep UV from 200-276 nm in channel 1, I am afraid the intensity in this channel will be very low. Through out the manuscript the data from the channels 1 and 2 are not shown nor used. Airborne measurements often face the problem of the instruments being too heavy, therefore I am surprised to read that the instruments has a channel 1 that seems not very useful. Maybe the authors can briefly comment about the potential use of the channels 1 and 2, or the former use in a different instrument. For channel 2 I can think of the retrieval of SO2 or HCHO, both would interesting for the presented study but may require a more detailed analysis.

The figure of the instrument (figure 1) is a bit confusing it might be clearer if the authors reduce the number of light beams. It seems that part of the "red" light beam originating from the "top" is blocked by the convex grating, I suppose this is not the case, may be because it is shifted relative to the drawing plane in the third dimension? Does second perspective view helps to explain more clearly? A radiometric calibration was performed as well as a spectral calibration. However, it seems the data form the radiometric calibration were not used it might be interesting to see the calibrated intensity in comparison to the LANDSAT 8 albedo (figure 7).

Parts of CCD are blocked to control offset and dark current which is a good idea if this part of the CCD can not be used for real observation. However in section 4 the well established pre-flight dark current and offset measurements are used and the dark measurements at the edges of the CCD are not mentioned.

Section 3 Research flight:

For non Chinese readers some more details about the measurement area might be nice to have. Can you add a map of China indicating where the city of Feiching is and in the context of the paper a satellite observation of NO2 for the day or season might be included as well.

Section 4 Data processing chain:
The authors use a wavelength range between 430 and 470 nm for the DOAS analysis including the cross sections of NO2, O3 and O4, as well as the ring cross section. However water vapour also shows some strong absorption lines in the respective wavelength range (Hitran data base, Rothman 2013), this is not included in the DOAS fit. (see general comment)

The tropospheric background of  $1 \times 10^{15}$  molec/cm2 as given by Popp et al. (2012) refers to the background around Zürich I am not sure this can be assumed for China as well. Here a satellite observation of the rural background around Feiching might help to estimate a realistic background.

The section about the Landsat 8 data analysis and resolution might be shifted from section 4.3.2 to 4.3.1. Meier et al. 2017 developed a method to retrieve the albedo from the measurements did you consider applying a similar method?

The assumption about the aerosol optical density, SSA and scattering function are partly given but not justified, the details about the optical density are not given directly.

During ascent and decent of the plane you can often estimate the PBL height from visibility or if available dew point measurements. This might be more realistic than the typical summer day assumption of 2000 m. However, the measurements were performed 2 years ago the respective information might be lost. The error is less than 13%, assuming that the PBL reached at least up to 1000 m.

The measurements by Tack et al. (2017) were performed in an SZA range of 40 to 60 degree; in this range the effect of the SZA might be different as in the range between 10 and 40 degree.

Section 6 discussion:

The uncertainty of the cross section is usually dominated by systematic uncertainty and not by the random errors. Only the random part is included in the QDOAS error analysis. AMTD
For the reference region a vertical column of  $1 \times 10^{15}$  molec/cm2 is assumed, therefore for the error of the slant column it should be multiplied by the respective AMF.

The focus of the paper is on the airborne instrument. A proper reference to the mobile DOAS instrument should be given, if this is not possible it might worth adding some additional information. Is the tropospheric or the total NO2 column retrieved? How large are the uncertainties of the mobile instrument? Are the same AMF settings (albedo, PBL height, aerosols) used as for the airborne observation?

Technical comments:

Please add an approximate scale in figure 2 or write some comparable information in the caption e.g. the measurement area was  $\sim 20 \times 30$  km and the distances between parallel lines were 1.5 km In figure 10 the individual sources are numbered it might help to add the numbers already in figure 2 in addition to the description.

L 130: replace one "across track" by "along track"

L 163: delete the last sentence starting with "the direct output..." this already written in I 152.

L 226: Is [28] a reference? Please use the AMT reference style.

L 320: "are near downwind of several plumes" replace by "inside the plumes" or "downwind of the sources"

Figure 8: "wavelength 20 nm" looks like a copy-paste-error from figure 9 "SZA 20°"?

Figures 10 and 11: please add the approximate position of the figure 11 in figure 10 and use the same numbers in figure 11.

References:

In some references there is a layout problem with subscripts like in NO2.

Rothman, L. S., Gordon, I. E., Babikov, Y., Barbe, A., Chris Benner, D., Bernath, P. F.,
Birk, M., Bizzocchi, L., Boudon, V., Brown, L. R., Campargue, A., Chance, K., Cohen, E. A., Coudert, L. H., Devi, V. M., Drouin, B. J., Fayt, A., Flaud, J.-M., Gamache, R. R., Harrison, J. J., Hartmann, J.-M., Hill C., Hodges, J. T., Jacquemart, D., Jolly, A., Lamouroux, J., Le Roy, R. J., Li, G., Long, D. A., Lyulin, O. M., Mackie, C. J., Massie, S. T., Mikhailenko, S., Müller, H. S. P., Naumenko, O. V., Nikitin, A. V., Orphal, J., Perevalov, V., Perrin, A., Polovtseva, E. R., Richard, C., Smith, M. A. H., Starikova, E., Sung, K., Tashkun, S., Tennyson, J., Toon, G. C., Tyuterev, VI. G., and Wagn, G.: The HITRAN 2012 molecular spectroscopic database, J. Quant. Spectrosc. Ra., 130, 4–50, https://doi.org/10.1016/j.jqsrt.2013.07.002, 2013.

---

## Author Response (AR1)

Authors' Response to Comments from anonymous referee #1

*Black, italic: Referee's comments*

Blue: Author's reply

Red: Changes in the original discussion paper

*This manuscript the first measurements from an airborne instrument, UVHIS, which measures backscattered light in the UV and visible parts of the electromagnetic spectrum. It includes instrument characteristics and background, data processing/calibration steps, and results from a NO2 VCD retrieval from its research flight near Feicheng, China. This paper fits the scope of AMT and would be a welcome read to the AMT audience for a new capability for high spatial*

10 *resolution trace gas observations in a new region of the world. However, before publishing, this manuscript requires the addressing of technical corrections that cause some concern, expansion of details in places, and some improvement on the quality of the writing. While most the comments are minor and related to the writing, I do recommend revisions in the major category because of the concern about the VCD measurement (which may be correct but needs more details to describe to convince the reader) and a great expansion of the mobile DOAS description. Detailed comments on those are below.*

First, we appreciate the overall positive response of the referee and we would like to thank for his constructive comments and helpful suggestions on the manuscript. As described below, we have modified the manuscript according to suggestions and provided clarifications where necessary.

20 *1. There are concerns about the actual calculation of the NO2 VCD as described in Sect 4.3. It is hard to tell what the VCD actually represents in the calculation. Is it a total column VCD? If so, the stratospheric details are missed. However, as it is stated that the stratospheric column is assumed stable during the flight from the reference and is canceled out leads the reader to believe it is not a total column. If it's just the below aircraft column, please state this and ensure the proper accounting for the distinction in the AMF and VCD calculation (e.g., Lamsal et al., 2017 is a great example that shows the*

25 *breakdown of how these components are calculated).*

In this manuscript, the $NO_2$ vertical columns we calculated are tropospheric vertical columns. We added necessary clarification throughout the manuscript. For example, we changed the title of the manuscript to 'The first high-resolution tropospheric NO2 observations from the Ultraviolet Visible Hyperspectral Imaing Spectrometer (UVHIS)'.

*2. Line 157-158: Pertaining to the spatial and temporal variability of the stratosphere being stable. This is maybe close to correct for a 3 hours flight, however there are changes in the SZA which will impact the slant column difference between the measurement and the reference. This could be estimated with a geometric calculation of the slant path with an assumed stratospheric amount between the reference and the measurement. It likely is small.*

According to TROPOMI L2 $NO_2$ product on 23 June, 2018, the $NO_2$ stratospheric vertical columns is about $3.5 \times 10^{15}$ molec cm$^{-2}$ in flight area. The SZA of reference spectra is about 13°, while the largest SZA of 37° occurred in the end of 3 hour flight. Under simple geometric approximation of light path, the largest difference of SCD of stratospheric $NO_2$ between the measurement and the reference is about $8 \times 10^{14}$ molec cm$^{-2}$.

*3. Lines 306 and 288: The mean slant column fitting error of 4.8x1015 molecules cm-2 and mean total value of error of 2.6x1015 molecules cm-2 for the VCD column with a range going down to 1x1015 molecules cm-2 does not seem to work out mathematically unless the AMF error is zero (which it is not) and the AMF must be ~2 (which is seems to be below that for most cases) and the error of the reference itself is 100% which is stated as 1x1015. Please check this math.*

We recalculated the $NO_2$ dSCD by adding a $H_2O$ vapor cross section from HITRAN database, the mean slant column fitting error slightly decreased (still about $4.8 \times 10^{15}$ molec cm$^{-2}$). We modified the tropospheric $NO_2$ vertical column of reference spectra to $3 \times 10^{15}$ molec cm$^{-2}$ with error of $1 \times 10^{15}$ molec cm$^{-2}$ based on TROPOMI $NO_2$ product. We checked the AMF calculation, and the mean value of AMF during the flight is about 2. We also update the figure of time series of AMF, surface reflectance, SZA and RAA. After the recalculation, the mean total value of error is $3.0 \times 10^{15}$ molec cm$^{-2}$ with a range from 1.5 to $5.9 \times 10^{15}$ molec cm$^{-2}$.

[Figure]

**Figure 7.** Time series of NO$_2$ AMF compared with **(a)** surface reflectance; **(b)** SZA and RAA for the research flight on 23 June 2018,
computed with SCIATRAN model based on the RTM parameters from the UVHIS instrument. Only data of the nadir observations in each
flight line are plotted.

*4. Line 17: the error of 2.6 x1015 is not the fitting error as stated. It is the error based on all sources of uncertainty.*

Corrected.

Measurements of nadir backscattered solar radiation of channel 3 are used to retrieve tropospheric vertical column densities
(VCDs) of NO$_2$ with a mean total error of $3.0 \times 10^{15}$ molec cm$^{-2}$.

*5. Are there literature references for the mobile-DOAS measurements? If not, then details on the specifics of that
measurement need to be greatly expanded upon as well as the zenith-sky NO2 retrieval in Sect 6.2. Especially details on the
uncertainty and what the VCD represented vertically (just the troposphere? Stratosphere?).*

We added two paragraphs in Sect 6.2 to describe the mobile DOAS tropospheric $NO_2$ retrieval method and its uncertainty analysis. For better comparison with UVHIS $NO_2$ observations, assumptions and parameters in $NO_2$ retrieval method for the mobile DOAS were set to the same as the UVHIS.

For better comparison with UVHIS $NO_2$ observations, assumptions and parameters in $NO_2$ retrieval method for the mobile DOAS were set to the same as the UVHIS. For example, residual amount of $NO_2$ in reference spectra was set to $3 \times 10^{15}$ molec cm$^{-2}$ with an error of $1 \times 10^{15}$ molec cm$^{-2}$; mobile DOAS observations only focus on tropospheric portion of $NO_2$ columns, assumed that the difference of the stratospheric $NO_2$ columns between observed spectra and reference spectra is negligible; vertical profiles of $NO_2$ and aerosol extinction, albedo, and aerosol properties in the AMF calculation were set to the same as UVHIS.

Like the uncertainty analysis of UVHIS $NO_2$ columns, the total uncertainty on the retrieved mobile tropospheric VCD is composed of three parts: (1) the mean uncertainty on dSCD of mobile DOAS is $1.4 \times 10^{15}$ molec cm$^{-2}$; (2) the uncertainty of reference vertical column is estimated to be $1 \times 10^{15}$ molec cm$^{-2}$. In the case that the tropospheric AMFs of measured and reference spectra are very close, this part results an uncertainty $1 \times 10^{15}$ molec cm$^{-2}$ to the total uncertainty; (3) the mean relative uncertainty on the AMF calculation is 22 % by square root of the quadratic sum of individual uncertainties like UVHIS. Combining these uncertainties together, the mean total uncertainties on the retrieved tropospheric $NO_2$ VCD is $2.1 \times 10^{15}$ molec cm$^{-2}$.

*6. In Figure 2, there are lines that are repeated in the northern half of the raster. Can this be described? Does this impact the comparisons to the mobile DOAS measurements? Please describe this overlap in the paper is this is what is shown in the NO2 data. The details should be discussed in the paper.*

Indeed, several fight lines are repeated in the northern part of flight area, but the spectral data of UVHIS are not recorded because of misoperation. Only spectral data of the flight lines over the steel factory are recorded, and the pass time of aircraft is 13:26 and 14:57 respectively. We added a paragraph and a figure to discuss this in detail. For impact on the comparison with mobile DOAS, see answer in question No. 22.

[Figure]

95

**Figure 12.** Three flight lines that pass through the steel factory, at local time 13:26 (a), 13:32 (c), and 14:57 (b). Panel (a) and (b) represent flight lines that cover the same area with a 1.5 hour time gap, panel (a) and (c) represent adjacent flight lines with a 6 minutes time gap.

Due to temporal discontinuity of flight lines and dynamic characteristics of $NO_2$ field, artefacts can be observed between

100    adjacent flight lines. Figure 12 shows three flight lines that pass through the steel factory, at local time 13:26 (a), 13:32 (c), and 14:57 (b). Panel (a) and (b) represent flight lines that cover the same area with a 1.5 hour time gap, panel (a) and (c) represent adjacent flight lines with a 6 minutes time gap. These flight lines can be divided into three regions: region A covers no $NO_2$ source but is affected by carbon factories about 3 km away; region B covers the steel factory as dominant $NO_2$ source; region C covers no $NO_2$ source and is not affected by other sources. Compared to region B, there is a large temporal

105    variety of $NO_2$ VCDs in region A between three flight lines. Region C is temporally consistent with relatively low $NO_2$

columns. From these observations it may be concluded that largest temporal variability could occur where there is no local NO$_2$ source but is down-wind of other sources, especially when wind direction is changing.

*7. Line 108: The mobile DOAS measurements are not shown in Figure 5 as stated. However, are shown in Figure 11.*
*Consider adding the location of these measurements in Figure 2 to show where the mobile DOAS measurements were taken.*
*Additionally, in Lines 288-290: technical details about mobile DOAS measurements are not mentioned before this. Discuss*
*these points within Sect 6.2.*

We added the location of mobile DOAS measurements in Figure 2, and added technical details about mobile DOAS VCD
retrieval method and its uncertainty analysis in Sect 6.2 as stated before.

[Figure]

**Figure 4.** Overview of the Feicheng demonstration flight on 23 June, 2018. Flight lines are shown in blue. Two orange circles represent the routes of mobile DOAS system. White dots numbered from 1 to 8 represent the major emission sources. Number 1: several carbon factories; number 2: a power plant; number 3-6: individual emitters inside the steel factories, while number 4 and 5 are inside the circle of one mobile DOAS route; number 7-8: two cement factories. White dashed box represents the reference area.

*8. Line 28: What is the intended meaning behind 'that NOx attracts large attention'. Please elaborate with some details and examples.*

125

Due to rapid industrialization and urbanization in the past few decades, China has become one of the largest $NO_x$ emitters in the world. As a result, China is experiencing a series of severe air pollution problems. Therefore, measuring $NO_x$ distribution by application of different techniques would benefit the pollutant emission detection and the air quality forecast. We modified this paragraph as below:

130

Nitrogen oxides ($NO_x$), the sum of nitrogen monoxide (NO) and nitrogen dioxide ($NO_2$), plays a key role in the chemistry of the atmosphere, such as the ozone destruction in the stratosphere (Solomon, 1999), and the secondary aerosol formation in the troposphere (Seinfeld and Pandis, 2016). In the troposphere, despite lightning, soil emissions and other natural processes, the main sources of NOx are anthropogenic activities like fossil fuel combustion by power plants, factories, and road

135 transportation, especially in the urban and polluted regions. As an indicator of anthropogenic pollution which leads to negative effects both on the environment and human health, the amounts and spatial distributions of $NO_x$ attract large attention. For example, China becomes one of the largest $NO_x$ emitters in the world due to fast industrialization, meanwhile China is also experiencing a series of severe air pollution problems in recent years (Crippa et al., 2018; An et al., 2019). Therefore measuring NOx distribution by application of different techniques, would benefit the pollutant emission detection

140 and the air quality trend forecast (Liu et al., 2017; Zhang et al., 2019).

*9. Line 54: add 'of NO2' after spatial distribution to clarify that this is the gas of interest in this paper.*

Corrected.

145

*10. Lines 67-68. Figure 1 only shows the optical bench for one of the channels and not all three as implied by the text. Please fix to the text saying that Figure 1 shows the optical bench for channel 3 and that the other two are similar.*

Corrected.

150

Figure 1 shows the optical bench of channel 3 and that the other two are similar. The optical design of each channel comprises a telecentric fore-optics, an Offner imaging spectrometer, and a two dimensional charge-coupled device (CCD) array detector.

155 *11. Line 86-87: reword this sentence to say that spectral and radiometric calibration in the laboratory were done prior to flights to reduce errors in spectral analysis. There shouldn't be a need to state it as 'very necessary'.*

Corrected.

160 *12. Line 38: Is this the first space-borne sensor ever in China or the first space-borne sensor related to air quality or trace gases?*

The EMI is the first space-borne sensor related to trace gas monitoring, we corrected this in the manuscript.

165 *13. Line 126: clouds were mentioned as filtered out. However, in the rest of the paper it says that the conditions were cloud free. Were there clouds to be filtered? If so, state where and how cloudy it was. If not, state that cloud filtering was not needed for these measurements due to clear skies. Same comment with the sun glint on water if applicable.*

The weather condition is cloud free on 23 June, 2018. However, sun glint on water occurred several times in the southern
170 part of flight area (especially over the river near the reference area) because of the low solar zenith angles. We add this statement in the manuscript.

The preprocessing procedure before spectral analysis includes data selection, dark current correction, spatial binning, and in-flight calibration. First, the spectral data acquired during U-turns of aircraft are removed in the processing because of the
175 large and changing orientation angles. Also a threshold of radiance values is set to neglect some over-illuminated ground pixels inside the flight area, which are usually caused by presence of cloud or water mirror reflection. During the entire flight, sun glint on water occurred several times in the southern part of flight area, especially above the river near the reference area. However, cloud was not present due to clean clear-sky weather condition.

180 *14. Line 165: Please revise to say something like 'and the properties that influence radiative transfer of light through the atmosphere' instead of 'and the radiative transfer'*

Corrected.

185 *15. Line 184: Please clarify which MODIS AOD product was used.*

MODIS AOD product used in this paper is MYD04 on 23 June, 2018, with resampling for every ground UVHIS pixel. We added this in the manuscript.

190 (5) Aerosol optical Depth (AOD) information used in AMF calculation is MODIS AOD product MYD04 at 470 nm on the same day with resampling for every ground UVHIS pixel (Remer et al., 2005), because neither ground-based aerosol measurement is performed, nor any AERONET station data near the flight area is are available.

*16. Line 184: What was the AOD measured from MODIS was during the flight? Please add this detail into the manuscript.*

195

MODIS AOD product used in this paper is MYD04 on 23 June, 2018, with resampling for every ground UVHIS pixel. We added this in the manuscript as stated before.

*17. Line 187: Please justify why SSA of 0.93 and asymmetry factor of 0.68 are used.*

200

The SSA and asymmetry factor of aerosol used in the manuscript are estimation of typical urban/industrial aerosols based on previous studies (Li et al., 2018).

Like the $NO_2$ profile, the aerosol extinction box profile is constructed from the PBL height and AOD. Single scattering
205 albedo (SSA) is assumed to be 0.93, and asymmetry factor is assumed to be 0.68 for aerosol extinction profile, based on previous studies of typical urban/industrial aerosols (Li et al., 2018).

*18. Line 226: Is [28] a referring to a reference? Please fix.*

210 Corrected.

*19. Consider consolidating Figures 6, 8, and 9 into one figure.*

We added an AMF dependency analysis on the VZAs, also a new panel in Figure 8. As shown in Fig. 8 (b) and (c), the
215 changes of AMF are 10% and 7% respectively, when other parameters are set as mean.

[Figure]

**Figure 8.** AMF dependence analysis results (a): on the surface reflectance; (b): on the SZAs; (c): on the VZAs; (d): on the wavelength.

20. Line 258: The difference in adjacent flight lines are not 'biases' but rather 'artifacts' of the changing NO2 VCDs due to temporal variation.

Corrected.

21. Section 6 would benefit from a more descriptive title, like 'NO2 VCD Assessment' rather than 'Discussion'

We changed the title of Sect 6 to 'NO$_2$ VCD Assessment'.

22. Lines 324-325. How do these results change if only considering points with a stricter temporal window between the mobile and aircraft measurements?

We added a new comparison to co-located mobile DOAS measurements only circled the steel factory, and the correlation coefficient improves to 0.86. In this case, all mobile measurements occurred inside the swath of one flight line of aircraft, and the time offset between instruments shortened to 15 minutes.

[Figure]

**Figure 14.** Scatter plot and linear regression analysis of the co-located $NO_2$ VCDs, retrieved from UVHIS and mobile DOAS system, (a) for all co-located measurements, (b) for co-located measurements that only circled the steel factory.

240 Figure 14 (a) shows scatter plots with VCDs retrieved by UVHIS on the x-axis and mobile DOAS VCDs on the y-axis, for all co-located measurements. The corresponding results of linear regression analysis are also provided in Fig.14 (a), with a correlation coefficient of 0.69, a slope of 1.30, and an intercept of-$9.01 \times 10^{14}$. The absolute time offset between mobile DOAS and airborne observations can be up to 1 hour, which means that both instruments cannot sample the $NO_2$ column at certain geolocation simultaneously. As shown in Fig. 14 (b), when only comparing UVHIS VCDs to mobile measurements

245 that circled the steel factory, the correlation coefficient improves to 0.86. In this case, all mobile measurements occurred inside the swath of one flight line of aircraft, and the time offset between instruments shortened to 15 minutes. In general, an underestimation of UVHIS VCDs of increased value can be observed in Fig 14 (a) and (b). Considering the variability in local emissions and meteorology, it is reasonable that the differences between these two instruments exist. Besides, the averaging effect of the area inside an UVHIS pixel can also lead to the underestimation of UVHIS compared to mobile

250 DOAS system.

*23. In line 323, the difference between the mobile DOAS measurements and the airborne measurements is described as an 'overestimate' of mobile DOAS measurements, but in the conclusions and abstract it is stated as an 'underestimate' by the aircraft. Please be consistent in this description in the manuscript.*

255

Corrected.

*24. Figure 2: the black dots are hard to see. Please change the color and/or symbol to make the points of interest stand out.*

260    We updated Figure 2 as stated before.

*25. Instead of having Table 2, could those results be translated into Figure 3(a) somehow? If keeping Table 2, then be more descriptive in the caption to say these are FWHMs at these wavelengths/angles.*

265    We added more information in the title of Table 2, also we added a new figure to plot the slit function shapes of 9 viewing angles at 450.504 nm.

**Table 2.** Preflight wavelength calibration results (FWHMs) of UVHIS channel 3 for 9 viewing angles. Light sources used in the calibration are a mercury-argon lamp and a tunable laser. Slit function shapes are retrieved by least square fitting of characteristic spectral lines, using a symmetric Gaussian function.

| FOV | 379.887 nm | 404.656 nm | 450.504 nm | 500.566 nm |
|---|---|---|---|---|
| -20° | 0.35 nm | 0.35 nm | 0.39 nm | 0.50 nm |
| -15° | 0.33 nm | 0.31 nm | 0.33 nm | 0.43 nm |
| -10° | 0.31 nm | 0.29 nm | 0.29 nm | 0.41 nm |
| -5° | 0.31 nm | 0.30 nm | 0.29 nm | 0.34 nm |
| 0° | 0.31 nm | 0.32 nm | 0.30 nm | 0.30 nm |
| 5° | 0.34 nm | 0.36 nm | 0.34 nm | 0.30 nm |
| 10° | 0.38 nm | 0.39 nm | 0.38 nm | 0.32 nm |
| 15° | 0.40 nm | 0.44 nm | 0.42 nm | 0.35 nm |
| 20° | 0.45 nm | 0.46 nm | 0.47 nm | 0.38 nm |

[Figure]

Figure 2. Measured slit functions (dots) at 450.504 nm and retrieved slit function shapes (lines) using a symmetric Gaussian function for 9 viewing angles.

275

*26. Lines 110-113: This text is redundant. These details were already stated in the previous paragraph.*

We reorganize this paragraph as below:

280    In the condition of spatial binning by 10 pixels across-track, the across-track spatial resolution of the ground pixel is about 22 m. At typical aircraft ground speed of 50 m/s and integration time of 0.5 s, the along-track spatial resolution of the ground pixel is about 25 m.

*27. Similarly, the first two paragraphs in Section 5 appear to be redundant. Please consolidate into one paragraph without*
285    *repeating details already stated.*

We reorganize Sect 5 as below:

The $NO_2$ tropospheric VCD two-dimensional distribution map is shown in Fig. 10 for the research flight on 23 June 2018.
290    With a high performance of UVHIS in spectral and spatial resolution, Figure 10 shows fine-scale $NO_2$ spatial variability to resolve individual emission sources. In general, the $NO_2$ distribution is dominated by several exhaust plumes with enhanced $NO_2$ concentration in the northwest part, which share a transportation pattern from south to north consistent with the wind direction. These sources include a power plant, a steel factory, two cement factories, and several carbon factories. The largest plume with peak values of up to $3 \times 10^{16}$ molec $cm^{-2}$, originates from an emitter inside a steel factory (number 3 in Fig. 10).
295    This dominant plume reaches its peak value outside at a small valley about 1 km north of the factory, and is transporting at least 9 km and seems to be continuing outside the flight region. This enhanced level of $NO_2$ may be caused by terrain factor which contributes to the accumulation of pollution gases.

Number 4 to 6 represent other emitters inside the steel factory. While the exhaust plumes originated from number 4 and 5 merge with the dominant plume, the plume from number 6 transports to north individually with a peak value of $1.4 \times 10^{16}$
300    molec $cm^{-2}$. A weaker plume with peak values of $1.5 \times 10^{16}$ molec $cm^{-2}$ is also detected by UVHIS, which seems to originate from the power plant. Indicated by number 2 in Fig. 10, this power plant is less than 2 km south of the steel factory. Number 1 in Fig. 10 indicates several carbon factories, which are located on the left side of the flight area. Several plumes with peak values of $1.5 \times 10^{16}$ molec $cm^{-2}$, gradually merge together during transportation downwind. Number 7 and Number 8 in Fig. 10 represent two different cement factories. Peak values of these two plumes are $1.5 \times 10^{16}$ molec $cm^{-2}$ and
305    $1.4 \times 10^{16}$ molec $cm^{-2}$ respectively.

Compared to the industrial areas mentioned above, the pollution levels of the rural areas are much lower due to the lack of contributing sources, ranging from 2 to $6 \times 10^{15}$ molec $cm^{-2}$. The urban area of Feicheng city is located on the right side of the flight area. Figure 11 is an enlarged map of UVHIS $NO_2$ observations over Feicheng city, with a color scale only extends to $7 \times 10^{15}$ molec $cm^{-2}$. Two black lines in Fig. 11 represent the truck roads in this city. The S104 is a provincial highway
310    that crosses Feicheng from north to south, while the S330 crosses Feicheng from east to west. Although lots of noise can be observed in Fig. 11, the $NO_2$ sources in Feicheng are mainly related to traffic and concentrated along the S104.

Due to temporal discontinuity of flight lines and dynamic characteristics of NO$_2$ field, artefacts can be observed between adjacent flight lines. Figure 12 shows three flight lines that pass through the steel factory, at local time 13:26 (a), 13:32 (c), and 14:57 (b). Panel (a) and (b) represent flight lines that cover the same area with a 1.5 hour time gap, panel (a) and (c) represent adjacent flight lines with a 6 minutes time gap. These flight lines can be divided into three regions: region A covers no NO$_2$ source but is affected by carbon factories about 3 km away; region B covers the steel factory as dominant NO$_2$ source; region C covers no NO$_2$ source and is not affected by other sources. Compared to region B, there is a large temporal variety of NO$_2$ VCDs in region A between three flight lines. Region C is temporally consistent with relatively low NO$_2$ columns. From these observations it may be concluded that largest temporal variability could occur where there is no local NO$_2$ sources but is down-wind of other sources, especially when wind direction is changing.

*28. There are grammar mistakes throughout the manuscript. These errors will need to be fixed before publication but I expect will be evolving in revisions. Some grammar and other writing fixes are located at the bottom of this review to help gives examples as to the types of errors found. They are not a full edit.*

Corrected.

*29. Consider a more concise title, such as, 'The first high-resolution NO2 observations from the Ultraviolet Visible Hyperspectral Imaging Spectrometer (UVHIS)'.*

We changed the title of the manuscript to 'The first high-resolution tropospheric NO$_2$ observations from the Ultraviolet Visible Hyperspectral Imaing Spectrometer (UVHIS)'.

*30. Does EMI capture this area? Or TROPOMI? It would be interesting to show some comparisons to those data products, especially since the flight was early afternoon on a cloud free day.*

Both EMI and TROPOMI capture this area on 23 June, 2018. In the figure below, we plot TROPOMI tropospheric NO$_2$ vertical columns because of its high spatial resolution. A quantitative comparison of the two retrievals may not make much sense because we use TROPOMI product to estimate the reference residual. However, enhancements in TROPOMI NO$_2$ are consistent with large UVHIS columns inside the flight area. Also, enhancements in TROPOMI NO$_2$ columns to the north of the flight area may indicate that the plumes originated from the steel factory keep transporting for tens of kilometers.

[Figure]

345  *31. What does Feicheng City look like if mapped on a color scale that only extends to 5x1015. Are there spatial patterns captured? It is hard to see any patterns in Figure 10 in that area due to the color scale expanding to much larger pollution scales. Perhaps a second panel in this figure would be interesting.*

We added a new figure of enlargement of UVHIS NO$_2$ VCD map over Feicheng city as below. More details can be found in answer to question No. 27.

[Figure]

**Figure 11.** Enlargement of UVHIS NO₂ VCD map over Feicheng city with a color scale only extends to $7 \times 10^{15}$ molec cm$^{-2}$. Two black lines in the map represent two truck roads that cross Feicheng city: S104, and S330.

*32. Figure 7: consider adding a true color image of this line to compare with the surface reflectance and AMF.*

For better comparison with surface reflectance and AMF, we added a panel in Figure 9 of radiance measurements from UVHIS. It is obvious that all panels share a same spatial distribution. However, some small differences can be observed due to time offset and spatial resolution difference.

[Figure]

Figure 9. (a) UVHIS Measured radiance; (b) Landsat 8 Surface reflectance; (c) computed AMFs, for one flight line of the Feicheng data set. A strong dependency of the AMF on the surface reflectance can be observed.

*33. Can you comment on applications of the other channels for UVHIS? Are there plans for other products in the future?*

The UVHIS is part of a full spectral multimodal airborne imaging spectrometer. Our works only involve the channel 2 and 3 for trace gas measurements. In future works, we will present the retrieval results of $SO_2$ or HCHO based on measurements of channel 2. For channel 1 that covers deep UV spectral range, it is beyond our scope of work, and it may be used for wildfire or other artificial UV light source detection.

Authors' Response to Comments from anonymous referee #2

*Black, italic: Referee's comments*

Blue: Author's reply

Red: Changes in the original discussion paper

*Xi et al. present a new airborne imaging DOAS instrument and results of the first demonstration flight. The results are encouraging and data might be interesting for further analysis such as satellite validation, emission estimates or model comparison. The paper fits very well in the context of AMT. However at some points more details might be required by the reader. Most of them are not critical but require an update of the manuscript.*

First, we appreciate the overall positive response of the referee and we would like to thank for his constructive comments and helpful suggestions on the manuscript. As described below, we have modified the manuscript according to suggestions and provided clarifications where necessary.

*1. Through out the manuscript the authors should take care to distinguish between total and tropospheric vertical columns - in most cases the tropospheric vertical column or to be more precise the column below the flight altitude is meant e.g. l 167.*

In this manuscript, the $NO_2$ vertical columns we calculated are tropospheric vertical columns. We added necessary clarification throughout the manuscript. For example, we changed the title of the manuscript to 'The first high-resolution tropospheric NO2 observations from the Ultraviolet Visible Hyperspectral Imaing Spectrometer (UVHIS)'.

*2. The NO2 fit shown in figure 4 has some residual structures, which might be noise but might also be caused by a systematic issue. The instrument was carefully calibrated before the measurements. The wavelength calibration is used only as apriori for the QDOAS software - which is certainly necessary. The slit function shape that can be extracted from the measurements using the Mercury-Argon lamp but are not shown or mentioned. Instead a symmetric Gaussian slit function is assumed in the DOAS analysis. The width of the slit function varies significantly within the fitting window (figure 3) - maybe the shape does so as well? I suggest adding at least a figure of the measured slit function for the extreme viewing directions (left, centre and right) including the Gaussian fit.*

We added a figure of measured slit functions at 450.504 nm for 9 viewing angles (-20°, -15°, -10°, -5°, 0°, 5°, 10°, 15°, 20°), and the respective Gaussian fit results. These Gaussian fit results suggest that a symmetric Gaussian function is a reasonable assumption for slit shape in all viewing directions.

[Figure]

Figure 2. Measured slit functions (dots) at 450.504 nm and retrieved slit function shapes (lines) using a symmetric Gaussian function for 9 viewing angles.

The preflight wavelength calibration was also performed in the laboratory, using a mercury–argon lamp and a tunable laser as light sources. We model the slit function of UVHIS using a symmetric Gaussian function. Spectral registration and slit function calibration are achieved by least square fitting of characteristic lines in collected spectra. Table 2 lists the retrieved full-width at half maximums (FWHMs) for channel 3. Figure 2 shows the measured slit functions at 450.504 nm for 9

viewing angles (-20°, -15°, -10°, -5°, 0°, 5°, 10°, 15°, 20°), and respective retrieved slit function shapes using a symmetric Gaussian function. These Gaussian fit results suggest that a symmetric Gaussian function is a reasonable assumption for slit shape in all viewing directions.

*3. The observations partly overlap as the distance between the parallel flightracks was 1.5 km and the swath width is 2.2 km. How good do the observed tropospheric NO2 columns agree in the overlapping regions, does this depend on the flight direction and time, according to figure 2 some parts of the flight track were covered at least twice.*

Indeed, several fight lines are repeated in the northern part of flight area, but the spectral data of UVHIS are not recorded because of misoperation. Only spectral data of the flight lines over the steel factory are recorded, and the pass time of aircraft is 13:26 and 14:57 respectively. We added a paragraph and a figure to discuss this in detail.

[Figure]

430

**Figure 12.** Three flight lines that pass through the steel factory, at local time 13:26 (a), 13:32 (c), and 14:57 (b). Panel (a) and (b) represent flight lines that cover the same area with a 1.5 hour time gap, panel (a) and (c) represent adjacent flight lines with a 6 minutes time gap.

Due to temporal discontinuity of flight lines and dynamic characteristics of $NO_2$ field, artefacts can be observed between

435  adjacent flight lines. Figure 12 shows three flight lines that pass through the steel factory, at local time 13:26 (a), 13:32 (c), and 14:57 (b). Panel (a) and (b) represent flight lines that cover the same area with a 1.5 hour time gap, panel (a) and (c) represent adjacent flight lines with a 6 minutes time gap. These flight lines can be divided into three regions: region A covers no $NO_2$ source but is affected by carbon factories about 3 km away; region B covers the steel factory as dominant $NO_2$ source; region C covers no $NO_2$ source and is not affected by other sources. Compared to region B, there is a large temporal

440  variety of $NO_2$ VCDs in region A between three flight lines. Region C is temporally consistent with relatively low $NO_2$

columns. From these observations it may be concluded that largest temporal variability could occur where there is no local NO$_2$ source but is down-wind of other sources, especially when wind direction is changing.

*4. The error given here is not the fitting error but the total error of the VCD - even if the fitting error ~4.8 x 10ˆ15 molec/cmˆ2 is probably the dominant contribution.*

Corrected.

*5. The instrument was build for airborne measurements in the troposphere, however the spectral range encompasses the deep UV from 200-276 nm in channel 1, I am afraid the intensity in this channel will be very low. Through out the manuscript the data from the channels 1 and 2 are not shown nor used. Airborne measurements often face the problem of the instruments being too heavy, therefore I am surprised to read that the instruments has a channel 1 that seems not very useful. Maybe the authors can briefly comment about the potential use of the channels 1 and 2, or the former use in a different instrument. For channel 2 I can think of the retrieval of SO2 or HCHO, both would interesting for the presented study but may require a more detailed analysis.*

The UVHIS is part of a full spectral multimodal airborne imaging spectrometer. Our works only involve the channel 2 and 3 for trace gas measurements. In future works, we will present the retrieval results of SO$_2$ or HCHO based on measurements of channel 2. For channel 1 that covers deep UV spectral range, it is beyond our scope of work, and it may be used for wildfire or other artificial UV light source detection.

*6. The figure of the instrument (figure 1) is a bit confusing it might be clearer if the authors reduce the number of light beams. It seems that part of the "red" light beam originating from the "top" is blocked by the convex grating, I suppose this is not the case, maybe because it is shifted relative to the drawing plane in the third dimension? Does second perspective view helps to explain more clearly? A radiometric calibration was performed as well as a spectral calibration. However, it seems the data form the radiometric calibration were not used it might be interesting to see the calibrated intensity in comparison to the LANDSAT 8 albedo (figure 7).*

We replot Figure 1 with less number of light beams in the manuscript, also another figure added here would help to explain the structure more clearly.
For better comparison with surface reflectance and AMF, we added a panel in Figure 9 of radiance measurements from UVHIS. It is obvious that all panels share a same spatial distribution. However, some small differences can be observed due to time offset and spatial resolution difference.

[Figure]

475

1. Telecentric fore-optics
2. Filter and slit
3. Concave mirror
4. Convex grating
5. CCD

Figure 1. Optical layout of the UVHIS channel 3. Optical design of channel 1 and channel 2 is similar.

[Figure]

Figure 9. (a) UVHIS Measured radiance; (b) Landsat 8 Surface reflectance, (c) computed AMFs, for one flight line of the Feicheng data set. A strong dependency of the AMF on the surface reflectance can be observed.

*7. Parts of CCD are blocked to control offset and dark current which is a good idea if this part of the CCD can not be used for real observation. However in section 4 the well established pre-flight dark current and offset measurements are used and the dark measurements at the edges of the CCD are not mentioned.*

Both pixels at left and right edge on the CCD detector are blocked to monitor dark current. During the flight, signals from these pixels are very stable due to the temperature stability of the instrument. For dark current correction, we prefer to use pre-flight dark current measurements for the entire CCD detector because the dark measurements at the edges of the CCD only cover a small part of the CCD.

*8. For non Chinese readers some more details about the measurement area might be nice to have. Can you add a map of China indicating where the city of Feiching is and in the context of the paper a satellite observation of NO2 for the day or season might be included as well.*

Feicheng is a county-level city in Shandong province, China, about 410 km away from Beijing. The flight area is located on the south bank of the Yellow River, at the western foot of Mount Tai. We added a figure of TROPOMI $NO_2$ tropospheric observation on 23 June, 2018, with the background Google map and the location of Feicheng.

[Figure]

Figure 3. TROPOMI observation of tropospheric NO$_2$ over China on 23 June, 2018. The location of UVHIS flight (Feicheng city) is also plotted in the map.

*9. The authors use a wavelength range between 430 and 470 nm for the DOAS analysis including the cross sections of NO2, O3 and O4, as well as the ring cross section. However water vapour also shows some strong absorption lines in the respective wavelength range (Hitran data base, Rothman 2013), this is not included in the DOAS fit. (see general comment)*

We recalculated the NO$_2$ dSCD by adding a H$_2$O vapor cross section from HITRAN database. The DOAS fit results and the errors both decreased slightly. For the fit example in the manuscript, the dSCD decreased from $5.05 \pm 0.38 \times 10^{16}$ molec cm$^{-2}$ to $4.95 \pm 0.34 \times 10^{16}$ molec cm$^{-2}$, while the RMS decreased from $4.56 \times 10^{-3}$ to $4.27 \times 10^{-3}$. We also updated the DOAS fit example figure.

[Figure]

**Figure 6.** Sample DOAS fit result for NO2: (**a**) observed (black dashed line) and fitted (blue line) optical depths from measured spectra; (**b**) the remaining residuals of DOAS fit.

515 *10. The tropospheric background of 1 x 10ˆ15 molec/cmˆ2 as given by Popp et al. (2012) refers to the background around Zürich I am not sure this can be assumed for China as well. Here a satellite observation of the rural background around Feiching might help to estimate a realistic background.*

We agree that a satellite observation of the rural area around Feicheng would be a more realistic background estimation. We
520 modified the tropospheric $NO_2$ vertical column of reference spectra to $3 \times 10^{15}$ molec cm$^{-2}$ with error of $1 \times 10^{15}$ molec cm$^{-2}$ based on TROPOMI $NO_2$ product on the same day. We also updated the UVHIS $NO_2$ distribution map.

[Figure]

**Figure 10.** Tropospheric $NO_2$ VCD map retrieved from UVHIS over Feicheng on 23 June 2018. The major contributing $NO_2$ emission sources are indicated by number 1 to 8.

*11. The section about the Landsat 8 data analysis and resolution might be shifted from section 4.3.2 to 4.3.1. Meier et al. 2017 developed a method to retrieve the albedo from the measurements did you consider applying a similar method?*

We shifted the section about the Landsat 8 data analysis and resolution from section 4.3.2 to 4.3.1. We realize that Meier et al. (2017) developed a method to retrieve surface reflectance from an instrument which is not radiometrically calibrated. However, surface reflectance algorithm for UVHIS is still under development and needs to be verified. We will include this in future works.

*12. The assumption about the aerosol optical density, SSA and scattering function are partly given but not justified, the details about the optical density are not given directly.*

MODIS AOD product used in this paper is MYD04 on 23 June, 2018, with resampling for every ground UVHIS pixel. The SSA and asymmetry factor of aerosol used in the manuscript are estimation of typical urban/industrial aerosol based on previous studies (Li et al., 2018).

(5) Aerosol optical Depth (AOD) information used in AMF calculation is MODIS AOD product MYD04 at 470 nm on the same day with resampling for every ground UVHIS pixel (Remer et al., 2005), because neither ground-based aerosol measurement is performed, nor any AERONET station data near the flight area is are available. Like the $NO_2$ profile, the aerosol extinction box profile is constructed from the PBL height and AOD. Single scattering albedo (SSA) is assumed to be 0.93, and asymmetry factor is assumed to be 0.68 for aerosol extinction profile, based on previous studies of typical urban/industrial aerosols (Li et al., 2018).

*13. During ascent and decent of the plane you can often estimate the PBL height from visibility or if available dew point measurements. This might be more realistic than the typical summer day assumption of 2000 m. However, the measurements were performed 2 years ago the respective information might be lost. The error is less than 13%, assuming that the PBL reached at least up to 1000 m.*

The PBL height was not measured during the flight on 23 June, 2018. Unfortunately, other respective information about PBL height was not available at present.

*14. The measurements by Tack et al. (2017) were performed in an SZA range of 40 to 60 degree; in this range the effect of the SZA might be different as in the range between 10 and 40 degree.*

We added an AMF dependency analysis on the VZAs, also a new panel in Figure 8. As shown in Fig. 8 (b) and (c), the changes of AMF are 10% and 7% respectively, when other parameters are set as mean.

[Figure]

**Figure 8.** AMF dependence analysis results (a): on the surface reflectance; (b): on the SZAs; (c): on the VZAs; (d): on the wavelength.

As can be seen in Fig. 7, the effect of sun and viewing geometries on AMFs is very small. Based on a previous study from Tack et al. (2017), changing SZA have the greatest effect on AMFs, in comparison to other sun and viewing geometries. In this study, we also did an AMF dependence analysis on SZAs and VZAs. The SZA varies from 12.8° to 37.4° during the 3 hour research flight, while the VZA ranges from 0° to 30° in most cases. As shown in Fig. 8 (b) and (c), the changes of AMF are less than 10% and 7% respectively, when other parameters are set as mean. Generally, a larger SZA or a larger VZA could result a longer light path through the atmosphere and thus a larger AMF.

*15. The uncertainty of the cross section is usually dominated by systematic uncertainty and not by the random errors. Only the random part is included in the QDOAS error analysis.*

We agree that the uncertainty of the cross section is usually dominated by systematic uncertainty. We added the word 'systematic' in this sentence.

*16. For the reference region a vertical column of 1 x10 ^15 molec/cm^2 is assumed, therefore for the error of the slant column it should be multiplied by the respective AMF.*

We modified the tropospheric $NO_2$ vertical column of reference spectra to $3 \times 10^{15}$ molec cm$^{-2}$ with error of $1 \times 10^{15}$ molec cm$^{-2}$ based on TROPOMI $NO_2$ product. Also, we fix this error in the calculation of error budget. The mean total value of error is $3.0 \times 10^{15}$ molec cm$^{-2}$ with a range from 1.5 to $5.9 \times 10^{15}$ molec cm$^{-2}$.

The second uncertainty source, $\sigma_{SCDref}$, is caused by the $NO_2$ residual amount in the reference spectra. Since we use TROPOMI tropospheric $NO_2$ product of the clean reference area as background amount, the uncertainty of $NO_2$ vertical column is estimated to be $1 \times 10^{15}$ molec cm$^{-2}$ directly from TROPOMI product. Assuming a tropospheric AMF of 2.0 and a tropospheric AMF over the reference spectra of 1.8, this results an uncertainty $9 \times 10^{14}$ molec cm$^{-2}$ to the tropospheric vertical column.

*17. The focus of the paper is on the airborne instrument. A proper reference to the mobile DOAS instrument should be given, if this is not possible it might worth adding some additional information. Is the tropospheric or the total NO2 column retrieved? How large are the uncertainties of the mobile instrument? Are the same AMF settings (albedo, PBL height, aerosols) used as for the airborne observation?*

We added two paragraphs in Sect 6.2 to describe the mobile DOAS tropospheric $NO_2$ retrieval method and its uncertainty analysis. For better comparison with UVHIS $NO_2$ observations, assumptions and parameters in $NO_2$ retrieval method for the mobile DOAS were set to the same as the UVHIS.

For better comparison with UVHIS $NO_2$ observations, assumptions and parameters in $NO_2$ retrieval method for the mobile DOAS were set to the same as the UVHIS. For example, residual amount of $NO_2$ in reference spectra was set to $3 \times 10^{15}$ molec cm$^{-2}$ with an error of $1 \times 10^{15}$ molec cm$^{-2}$; mobile DOAS observations only focus on tropospheric portion of $NO_2$ columns, assumed that the difference of the stratospheric $NO_2$ columns between observed spectra and reference spectra is negligible; vertical profiles of $NO_2$ and aerosol extinction, albedo, and aerosol properties in the AMF calculation were set to the same as UVHIS.

Like the uncertainty analysis of UVHIS $NO_2$ columns, the total uncertainty on the retrieved mobile tropospheric VCD is composed of three parts: (1) the mean uncertainty on dSCD of mobile DOAS is $1.4 \times 10^{15}$ molec cm$^{-2}$; (2) the uncertainty of reference vertical column is estimated to be $1 \times 10^{15}$ molec cm$^{-2}$. In the case that the tropospheric AMFs of measured and reference spectra are very close, this part results an uncertainty $1 \times 10^{15}$ molec cm$^{-2}$ to the total uncertainty; (3) the mean relative uncertainty on the AMF calculation is 22 % by square root of the quadratic sum of individual uncertainties like UVHIS. Combining these uncertainties together, the mean total uncertainties on the retrieved tropospheric $NO_2$ VCDs is $2.1 \times 10^{15}$ molec cm$^{-2}$.

*18. Please add an approximate scale in figure 2 or write some comparable information in the caption e.g. the measurement area was ~20 x 30 km and the distances between parallel lines were 1.5 km In figure 10 the individual sources are numbered it might help to add the numbers already in figure 2 in addition to the description.*

We updated this figure as below:

[Figure]

620

**Figure 4.** Overview of the Feicheng demonstration flight on 23 June, 2018. Flight lines are shown in blue. Two orange circles represent the routes of mobile DOAS system. White dots numbered from 1 to 8 represent the major emission sources. Number 1: several carbon factories; number 2: a power plant; number 3-6: individual emitters inside the steel factories, while number 4 and 5 are inside the circle of one mobile DOAS route; number 7-8: two cement factories. White dashed box represents the reference area.

625

*19. L 130: replace one "across track" by "along track"*

Corrected.

630  *20. L 163: delete the last sentence starting with "the direct output..." this already written in l 152.*

Corrected.

*21. L 226: Is [28] a reference? Please use the AMT reference style.*

635

Corrected.

*22. L 320: "are near downwind of several plumes" replace by "inside the plumes" or "down-wind of the sources"*

640

Corrected.

*23. Figure 8: "wavelength 20 nm" looks like a copy-paste-error from figure 9 "SZA 20∘"?*

645

We consolidated Figures 6, 8, and 9 into new Fgure 8, also we corrected this error.

*24. Figures 10 and 11: please add the approximate position of the figure 11 in figure 10 and use the same numbers in figure 11.*

650

We updated Figure 11 with same numbers as below:

[Figure]

**Figure 13.** Overview of VCDs retrieved from ground-based mobile DOAS system (circle marks), and VCDs retrieved by UVHIS (background layer), measured on 23 June 2018.

655

*25. In some references there is a layout problem with subscripts like in NO2.*

Corrected.

[revised manuscript text omitted]

---

## Author Response (AR3)

Authors' response to comments and suggestions from referees and the editor

*Black, italic: Referee's comments*

Blue: Author's reply

Red: Changes in the original discussion paper

*1. in the manuscript, you state in line 289 that "the NO2 sources in Feicheng are mainly related to traffic and concentrated along the S104". I frankly cannot see that in the figure and suggest to remove this statement.*

Removed.

*2. in the caption of Figure 10, you write "NO2 emission". Please replace by "NOx emission" as emissions are usually in the form of NO, not NO2.*

Replaced NO2 emission with NOx emission.

*3. when introducing the car-DOAS measurements, please add that the retrieval window in the UV differs from the one used for the airborne observations.*

We added this statement in line 338.

It is worth noting that the retrieval window in the mobile DOAS observations differs from the one used for the airbrine observations.

*4. please add acknowledgements and data sources for TROPOMI and LANDSAT 8 data.*

We added acknowledgements and data sources for TROPOMI and LANDSAT 8 data.

*Acknowledgments*. We would like to thank Thomas Danckaert, Caroline Fayt and Michel Van Roozendael for help on QDOAS software. We are thankful to the following agencies for providing the satellite data: The Sentinel 5 Precursor TROPOMI Level 2 $NO_2$ product is developed by KNMI with funding from the Netherlands Space Office (NSO) and processed with funding from the European Space Agency (ESA). TROPOMI data can be downloaded from https://s5phub.copernicus.eu. Landsat 8 OLI data have been produced, archived, and distributed by the U.S. Geological Survey (USGS). The original Landsat surface reflectance algorithm was developed by Dr. Eric Vermote, NASA Goddard Space Flight Center (GSFC). Landsat 8 OLI data are available at https://earthexplorer.usgs.gov/.

*5. in the introduction, it would be good to also make reference to some other imaging DOAS instruments that have been flown for NO2 measurements:*

We added these references of other imaging DOAS instruments in section 1.

For the purpose of retrieval of urban $NO_2$ horizontal distribution, Popp et al. (2012), General et al. (2014), Schönhardt et al. (2015), Lawrence et al. (2015), Nowlan et al. (2016), and Lamsal et al. (2017) performed their measurements separately in Zürich, Switzerland, Indianapolis and Barrow, USA, Ibbenbüren, Germany, Leicester, England, Houston, USA, and Maryland, USA.

45

Lamsal, L. N., S. J. Janz, N. A. Krotkov,K. E. Pickering, R. J. D. Spurr,M. G. Kowalewski, C. P. Loughner,J. H. Crawford, W. H. Swartz, and J. R. Herman (2017), High-resolution $NO_2$ observations from the Airborne Compact Atmospheric Mapper: Retrieval and validation, J. Geophys. Res.Atmos., 122, 1953–1970, doi:10.1002/2016JD025483.

50

Nowlan, C. R., Liu, X., Leitch, J. W., Chance, K., González Abad, G., Liu, C., Zoogman, P., Cole, J., Delker, T., Good, W., Murcray, F., Ruppert, L., Soo, D., Follette-Cook, M. B., Janz, S. J., Kowalewski, M. G., Loughner, C. P., Pickering, K. E., Herman, J. R., Beaver, M. R., Long, R. W., Szykman, J. J., Judd, L. M., Kelley, P., Luke, W. T., Ren, X., and Al-Saadi, J. A.: Nitrogen dioxide observations from the Geostationary Trace gas and Aerosol Sensor Optimization (GeoTASO) airborne instrument: Retrieval algorithm and measurements during DISCOVER-AQ Texas 2013, Atmos. Meas. Tech., 9, 2647–2668, https://doi.org/10.5194/amt-9-2647-2016, 2016.

55

Schönhardt, A., Altube, P., Gerilowski, K., Krautwurst, S., Hartmann, J., Meier, A. C., Richter, A., and Burrows, J. P.: A wide field-of-view imaging DOAS instrument for two-dimensional trace gas mapping from aircraft, Atmos. Meas. Tech., 8, 5113-5131, doi:10.5194/amt-8-5113-2015, 2015.

60

General, S., Pöhler, D., Sihler, H., Bobrowski, N., Frieß, U., Zielcke, J., Horbanski, M., Shepson, P. B., Stirm, B. H., Simpson, W. R., Weber, K., Fischer, C. and Platt, U.: The Heidelberg Airborne Imaging DOAS Instrument (HAIDI) – a novel imaging DOAS device for 2-D and 3-D imaging of trace gases and aerosols, Atmos. Meas. Tech., 7(10), 3459–3485, doi:10.5194/amt-7-3459-2014, 2014.

65

*6. Line 133: Can the authors quantify in the text the radiance threshold applied?*

We used a radiance threshold of $12.8\ \mu W \cdot cm^{-2} \cdot sr^{-1} \cdot nm^{-1}$ at 450 nm.

Furthermore, a radiance threshold of $12.8\ \mu W \cdot cm^{-2} \cdot sr^{-1} \cdot nm^{-1}$ at 450 nm is set to neglect some over-illuminated ground pixels inside the flight area, which are usually caused by the presence of clouds or water mirror reflection.

*7. Equation 1 is valid for a total VCD, but not tropospheric VCD as is referred to below this equation. .The stratospheric slant column must also be subtracted from the dSCD. In lines Line 167-170, it is stated that the stratospheric column cancels out during flight from the reference but in the comment back to the reviewers it was quantified as a difference up to 8e14 due to the changes in the slant path through the atmosphere, which is not stated in the text. This is an impact and should either be added to this product or by quantified in the error estimate. It is a change of 23% over the time period of the flight...*

We added this stratospheric correction in Sect 4.2.

Changes in the stratospheric $NO_2$ could also propagate to the measured tropospheric columns of UVHIS. Under the assumption of a constant stratosphere in time and space during the flight, the changes in the SZA impact the column difference between the measurement and the reference. To correct the change in the stratospheric $NO_2$ SCD, we apply a geometric approximation of the stratospheric AMF with a stratospheric VCD of $3.5 \times 10^{15}$ molec $cm^{-2}$ from TROPOMI product. The maximum change in the stratospheric SCD with respect to the reference, was $8 \times 10^{14}$ molec $cm^{-2}$.

*8. Line 195: State the range of AOD measurements from MODIS for this flight day.*

The MODIS AODs in the flight area range from 0.142 to 0.365.

*9. Line 228: Inconsistencies are expected because one is radiance and the other is surface reflectance. They aren't the same thing thought they should be similar in spatial pattern. I don't see differences that would be caused by time differences with how it is displayed here. The one detail that does stick out though is that roadway isn't visible in the radiance measurements even though UVHIS should be higher resolution than Landsat, so can this be explained just by the color bar range choice? Perhaps but the unweighted averaging within the grid as discussed in Section 4.4? It would be nice to have a sentence or two about this.*

Actually, the roadway is visible in Fig. 9 (a), but the width of the road looks thinner compared to Fig. 9 (b). We replaced this figure with a higher resolution one. We modified the last sentence in this paragraph.

Fig. 9 also shows several slight inconsistencies between the UVHIS measured radiance and the Landsat 8 surface reflectance product. For example, the east-west main road looks thinner in Fig. 7 (a) compared to Figs. 7 (b) and (c). This could be explained by the relatively higher spatial resolution performance of the UVHIS and the resampling of Landsat 8 pixels.

*10. Line 278: The authors refer to this 'weaker' plume, but it has a higher column density than the last described plume.*

We deleted the word 'weaker'.

*11. Is the noise in Figure 11 primarily attributable to surface reflectivity?*

We thought the noise in Fig. 11 is primarily attributable to random noise, because the spatial patterns of VCD and surface reflectance (or radiance) are inconsistent.

*12. Line 297: This is not correct. Areas with NO2 sources can vary dramatically with time. This is a sample size of 2. Emissions from sources aren't always constant and their footprints can change dramatically due to meteorology as well. In the case of region B, the variability is almost just as large as it is in region a (maybe even more so). The only consistent region is region C.*

We agreed this correction and modified the last few sentences in this paragraph.

In these three regions, only region C is temporally consistent with relatively low $NO_2$ columns, whilst a large temporal variety of $NO_2$ VCDs exists in region A and region B because of inconstant emission sources and changing meteorology.

*13. Line 315: The authors refer to AMF of 2 and AMF of 1.8 for the reference in the same sentence for the reference. This is confusing. I think we need the AMF for TROPOMI in this case to convert 1e15 VCD to its SCD then divide by the AMF from UVHIS.*

We thought the TROPOMI AMF of the reference may be inappropriate considering the different viewing angles. The AMF of 1.8 is near the mean value of the UVHIS AMFs in the reference area, and the AMF of 2.0 is near the mean value of all UVHIS AMFs.

135    *14. Line 365: This comment is in relation to the low bias by UVHIS and high bias by mobile DOAS. It seems that the mobile DOAS would probably be more sensitive to near surface pollution than the aircraft. Is the assumption of the a priori profile a reason for the aircraft being lower than the mobile DOAS and is there an optimal profile between the two that would result in the two being closer to 1:1. I am not suggesting to redo all work with a new a priori but perhaps doing the sensitivity test like was done on the airborne work on the mobile DOAS and see if that is a plausible cause for the biases. Then reflect on*
140    *that in the text.*

We did a sensitivity test, the detail is in next answer.

*15. Last sentence in Section 6: We haven't been given reason to believe that the spatial footprint of the mobile DOAS is*
145    *smaller than the UVHIS pixels. Therefore, this sentence is not a valid reason for these results. The above comment seems more plausible.*

We deleted this sentence and modified to

150    A sensitivity test of the AMF on the $NO_2$ profile was performed for all co-located measurements, using a box profile of 500 m. Compared to the box profile of 2 km, the UVHIS AMFs decreased by an average of 17 %, whilst the mobile DOAS AMFS decreased by an average of 2.7 %. This results suggest that a more realistic profile with the $NO_2$ layer closer to the ground could improve the slope closer to unity.

155    *16. Line 20 + final paragraph in Section 6: It is nice to see this improvement in the correlation between Fig 14 a and b. Is the correlation different because of the time difference between measurements or because of the location near the steel factory? It seems more likely to focus on is the difference in time between measurements in the text rather than focusing on the location near the steel plant. This should be detailed in the caption of Figure 14 as well.*

160    We added the time difference information in the caption of Fig. 14.

*17. Title has a misspelled word: Imaing = imaging*

Fixed
165
*18. Line 44: Change 'lower' to 'coarser'*

Fixed

170   *19. Delete Lines 120-122. These details are on the next page and fit better there.*

Deleted

*20. Line 215: Delete Other RTM parameters used in the AMF calculations are also provided in Fig. 7. Because there are no*
175   *other parameters shown that I can see.*

Deleted

*21. Line 236: 'estimated' PBL height.*

180
Fixed

*22. Should section 4.4 go before 4.3 or even before 4.1 because geolocation and other steps would have to be performed*
*before AMF analysis to link it to surface reflectivity and other attitude parameters?*
185
We added a geo-referencing part in Sect 4.1 and changed the title of Sect 4.4 to 'Resampling and mapping'.

*23. Line 264: refer to Figure 10.*

190   We added 'As shown in Fig. 10' in this sentence.

*24. Line 310: the mobile DOAS system hasn't been introduced yet so it seems inappropriate to bring up here.*

Deleted
195
*25. Table 3 and Table 5: In the text, please explain what the intensity offset is.*

This is the offset parameter in the DOAS fitting for possible instrumental and/or atmospheric stray light or residual dark
current signal. We changed the 'Intensity offset' to 'Offset' to reduce misunderstandings.

200

*26. Figure 4, 10, 12, and 13: Please make the numbers indicating the point sources larger. They're hard to read.*

We updated these figures.

205 *27. The authors included water vapour in the DOAS analysis, unfortunately this did not improve the fit. Nevertheless the water cross section should be added in Table 5.*

In the manuscript, Table 5 lists the parameters of mobile DOAS system. We did not add a water vapour cross section because the absorption of water vapour is weak in the 356-376 nm fit window of the mobile DOAS.

210

215

220

225

230

Relevant changes:

235

1. Modified the title.

2. Modified the affiliation.

3. Added a stratospheric correction in the DOAS analysis.

[revised manuscript text omitted]